# Operational and experimental snow observation systems in the upper Rofental: data from 2017 to 2023

Michael Warscher[1], Thomas Marke[1], Erwin Rottler[1], and Ulrich Strasser[1]

[1]Department of Geography, University of Innsbruck, 6020 Innsbruck, Austria

**Correspondence:** Michael Warscher (michael.warscher@uibk.ac.at)

**Abstract.** This publication presents a comprehensive hydrometeorological data set for three research sites in the upper Rofental (1891–3772 m a.s.l., Ötztal Alps, Austria) and is a companion publication to a data collection published in 2018. The time series presented here comprise data from 2017 to 2023 and originate from three meteorological and snow-hydrological stations at 2737, 2805, and 2919 m a.s.l. The fully equipped automatic weather stations include a specific set of sensors to continuously record snow cover properties. These are automatic measurements of snow depth, snow water equivalent, volumetric solid and liquid water content, snow density, layered snow temperature profiles, and snow surface temperature. One station is extended by a particular arrangement of two snow depth and water equivalent recording devices to observe and quantify wind-driven snow transport. These devices are installed at nearby wind-exposed and sheltered locations and are complemented by an acoustic-based snow drift sensor. We present data for temperature, precipitation, humidity, wind speed, and radiation fluxes and explore the continuous snow measurements by combined analyses of meteorological and snow data to show typical seasonal snow cover characteristics. The potential of the snow drift observations is demonstrated with examples of measured wind speeds, snow drift rates and redistributed snow amounts during several blowing snow events. The data complement the scientific monitoring infrastructure in the research catchment and represent a unique time series of high-altitude mountain weather and snow observations. They enable comprehensive insights into the dynamics of high altitude meteorological and snow processes and are collected to support the scientific community, local stakeholders and interested public, as well as operational warning and forecasting services.

## 1 Introduction

Mountain regions are subject to particularly fast environmental changes induced by the rapid development of changing global climate. They are likely to be more vulnerable in the expected consequences on the typical mountain ecosystems (Beniston et al., 2018; Pörtner et al., 2019). Much evidence has been collected and documented in the recent past showing that the rate of climate change induced temperature change varies with elevation (Ohmura, 2012; Pepin et al., 2015; Wang et al., 2018; Pepin et al., 2022). Recent research focuses on impacts on precipitation amounts (and rain/snowfall rates), glacial ice losses, changes in snow cover and melt dynamics, and consequent runoff behavior and water supply in mountain regions (e.g., Barnett et al., 2005; Trenberth, 2011; Beniston et al., 2018; Blöschl et al., 2019). The spatial patterns in the changes in high mountain snow cover, their implications for water storage and release, and their function as protection sheet for glaciers in a changing climate

are still poorly understood. E.g., Musselmann et al. (2017) and Wu et al. (2018) found the counterintuitive effect of slower snowmelt in a warming climate due to the seasonally earlier melting period in the future with less radiative energy input. All these open research questions underline the specific importance of high altitude observations of snow and climate because data in these regions is still sparse compared to lower elevations, even in a well monitored and investigated mountain range as the European Alps (Matiu et al., 2021).

The Rofental in the Ötztal Alps has developed into a well-recognized high Alpine environmental research basin and cooperation platform. Strasser et al. (2018) documented the history of diverse environmental observation data for the Rofental going back to the year 1850. In this original publication a comprehensive collection of the available data sets are compiled for the Rofental until 2017. These are accessible at https://doi.pangaea.de/10.1594/PANGAEA.876120. The data documented by Strasser et al. (2018) has been extensively used to investigate snow-hydrological, glaciological and meteorological processes. E.g., Klug et al. (2018) summarized annual mass balances from 2001 to 2011 for the Hintereisferner (HEF) by combining geodetic and airborne laserscanning data. Rieg et al. (2018) presented the applicability of Pléiades tri-stereo satellite data to derive multi-temporal high-resolution DEMs, and calculated mass balances for the Hochjochferner. Schmieder et al. (2018) used the data for hydrograph separation in a stable water isotope measurement campaign. Glaciological, hydrological, and meteorological data was used by Hanzer et al. (2016, 2018) to validate results of a physically-based snow-hydrological model, and to assess climate change impacts on the hydrological system of the catchment. De Gregorio et al. (2019a, b) used data from the catchment to develop and evaluate novel remote sensing techniques for satellite-based retrieval of snow coverage and SWE. A similar study was conducted by Rastner et al. (2019) who focused on automated mapping of snow cover and snow line altitudes from Landsat data. Zolles et al. (2019) used the data to present an uncertainty assessment of glacier mass and energy balance models, Stoll et al. (2020) to compare the impact of different model approaches on glacio-hydrological climate change studies in high-mountain catchments. The effect of spatial and temporal flow variations on turbulent heat exchange at the Hintereisferner was investigated by Mott et al. (2020). The Hintereisferner was recently subject to several intense measurement and modeling campaigns (Goger et al., 2022; Voordendag et al., 2023a, b, c). Schmidt et al. (2023) reconstructed sediment transport from the catchment using meteorological and discharge data. Strasser et al. (2024) used the data to demonstrate and evaluate the open source snow-hydrological model openAMUNDSEN that is optimised for mountain catchments.

Much of the measurement infrastructure documented in Strasser et al. (2018) has been further maintained without changes. Some of the installations were modified and modernized to meet current technical standards (mostly the ones of operational data transmission). Most important however, the existing observation network has been extended with a new automatic weather station, and it has been complemented by sensors continuously recording snow cover properties. In the following, we document these extensions of the observation programme and present the new data sets that have been recorded between 2017 and August 2023. Our data from the high Alpine glaciated catchment of the Rofental complement measurements from sites in other climates and/or geographical settings: comparable data sets with meteorological and snow cover observations spanning a longer time period for single stations were published by Marty et al. (2012) for Weissfluhjoch, Switzerland, 2540 m a.s.l., and by Morin et al. (2012) for Col de Porte, France, 1325 m a.s.l. (extended by Lejeune et al., 2019). Sicart et al. (2023) recently complemented the Col de Porte site with data from a field campaign focused on snow-forest interaction. Another similar data collection was

published by Pradhananga et al. (2021) for the Peyto Glacier Research Basin (Canada). Lundquist et al. (2024) presented and analysed data from an intense measurement campaign over one winter season in the East River Watershed, Colorado (USA). Ménard et al. (2019) compiled several station data sets from different cold region sites for snow model comparison.

We show the potential of the new data and specifically the different snow monitoring sensors by presenting exemplary use cases of data analysis. The new data time series are extending the continuous meteorological and snow observations in the upper Rofental to support I) improved process understanding of snow drift, accumulation and melt dynamics in high mountain regions, II) process model development, evaluation, and application on different scales and for different purposes (e.g., regional climate and weather, glaciology, hydrology, ecology) and III) the operational avalanche warning and flood forecasting services of Tyrol (Austria) and South Tyrol (Italy) by providing real-time observations.

## 2   The Rofental - site description

The Rofental (98.1 km$^2$, Fig. 1) is a glaciated headwater catchment in the Central Eastern Alps, namely in the upper Ötztal Alps (Tyrol, Austria). It is described in detail in the first data publication by Strasser et al. (2018). Here, we give a short summary of the site description. The Rofental stretches from 1891 m a.s.l. at the gauge at Vent, the lowest point of the catchment, to 3772 m a.s.l. at the top of Wildspitze, the highest summit of Tyrol. It is characterized by a valley floor in the lower part which is a narrow discontinuous riparian zone typically less than 100 m in width. The average slope of the catchment is 25° and the average elevation is 2930 m a.s.l. The Rofenache river is a tributary to the Venter Ache, Ötztaler Ache and the Inn and as such contributing to the Danube system (i.e., the Black Sea). The runoff regime of the Rofenache is of glacial to nivo-glacial type and has not been modified by any measures of hydropower generation. It is dominated by the melt of snow and ice during spring and summer, respectively, and the early melt season onset is typically in April. The climate of the Rofental is characterized as an inner Alpine dry type. Due to its complex topography, the Rofental is characterized by steep environmental gradients and large spatio-temporal variations of the meteorological conditions. The mean annual temperature at the station in Vent (1900 m a.s.l., 46.85833°N, 10.91250°E) is 2.5 °C, and total annual precipitation varies between 797 mm in Vent (1982–2003, Kuhn et al., 2006) and around 1500 mm in the higher altitudes at 3000 m a.s.l. In these higher regions, seasonal snow cover lasts from October until the end of June (Strasser et al., 2018). Land cover in the Rofental is dominated by mountain pastures and natural grasslands in the lower areas, and sparsely vegetated areas as well as bare rocks and glaciers in the higher elevations (Fig. 1). Maintenance of the monitoring instrumentation and fieldwork in the valley is supported by the given accessibility - partly also in winter - and logistical infrastructure. A research station at Hintereisferner (HEF, built in 1966 in 3026 m a.s.l.) and one at Vernagtbach (built in 1973 in 2637 m a.s.l.) serve as logistic bases for fieldwork on the two glaciers. Several mountain huts are located in the Rofental, namely the "Vernagthütte/Würzburger Haus" (2755 m a.s.l.), the "Hochjoch-Hospiz" (2413 m a.s.l.), the "Brandenburger Haus" (3277 m a.s.l.) and close by the Austrian-Italian borderline at the Hochjoch the "Schöne Aussicht Schutzhütte" ("Rifugio Bella Vista", 2845 m a.s.l.), in the Schnalstal glacier ski resort. The "Rofenhöfe" (2014 m a.s.l.), the highest permanently settled mountain farm in Austria, are well situated as base camp in the lower valley floor, accessible throughout the year.

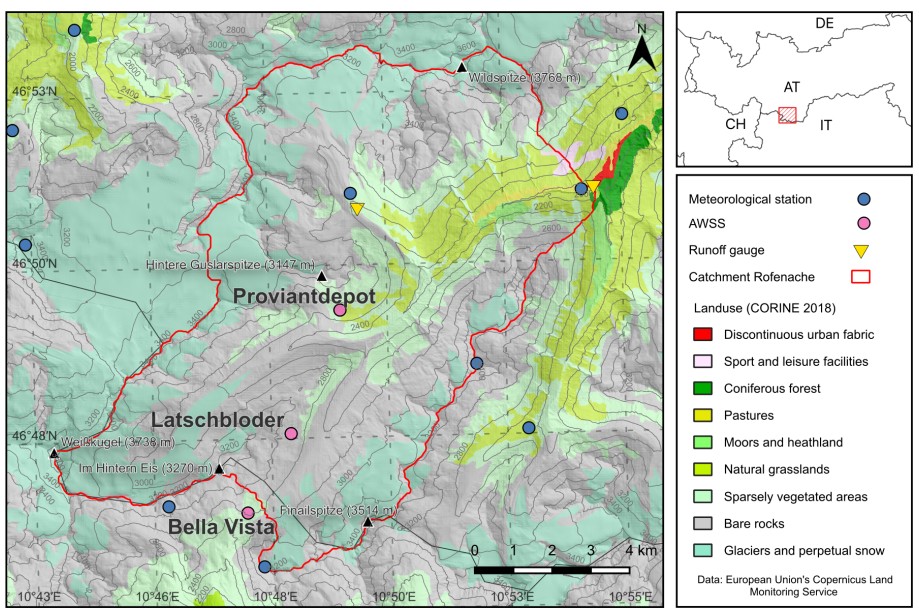

**Figure 1.** The research basin Rofental and the Rofenache catchment ($98.1\,\text{km}^2$) with the three automatic weather and snow stations Bella Vista ($2805\,\text{m a.s.l.}$), Latschbloder ($2919\,\text{m a.s.l.}$), and Proviantdepot ($2737\,\text{m a.s.l.}$) are highlighted. A map displaying all surrounding monitoring stations can be found in Strasser et al. (2018). The land cover data is the CORINE Land Cover 2018 data set (https://doi.org/10.2909/71c95a07-e296-44fc-b22b-415f42acfdf0).

## 3   The observation network and its development

Apart from its history within UNESCO IHP (https://en.unesco.org/themes/water-security/hydrology, last access: June 13, 2024), the Rofental catchment is part of several international research initiatives. It is a research basin in the framework of the GEWEX INARCH project (https://inarch.usask.ca/, last access: June 13, 2024) and part of the ERB Euro-Mediterranean Network of Experimental and Representative Basins (https://erb-network.simdif.com, last access: June 13, 2024). Further, it is a regular complex site in the LTSER platform Tyrolean Alps (https://lter-austria.at/ta-tyrolean-alps, last access: June 13, 2024) which belongs to the national and international long term ecological research networks LTER Austria, LTER Europe and ILTER (see e.g., Ohl et al., 2010; Angelstam et al., 2019). The Hintereisferner station is part of the EU Horizon 2020 INTERACT framework of Arctic (and a few Alpine) research stations (https://eu-interact.org/field-sites/station-hintereis/, last access: June 13, 2024).

An ongoing effort has been undertaken in the past years to supplement data available from the lower regions with additional automatic weather station (AWS) installations in the higher elevations. Since the reported state of the technical instruments in the catchment in Strasser et al. (2018), two of the eight existing weather stations have been extended and modified, and a new AWS has been brought into operation in 2019. We here present the measurement systems and the data from these three stations which are the locations for the extensive additional snow observations.

## 4 Meteorological stations and automatic snow cover measurements

In the uppermost parts of the Hochjoch valley, two AWSs have been brought into operation in 2013 and 2015: Latschbloder (2919 m a.s.l.) in September 2013, and Bella Vista (2805 m a.s.l.) in June 2015. To complement the station network, a new AWS has been installed in September 2019: the Proviantdepot station (2737 m a.s.l.). For all stations, the height of the sensors above ground is at least 2 m; in winter, the distance between the snow surface and the sensors can become much smaller, and in extreme snow-rich periods the instruments even can become completely snow-covered. Such periods can be recognized in the data by typical recordings of zero wind speed and increasing dampening of the other meteorological variables.

An important aim in the conceptual development of the Rofental measurement network was - in addition to the meteorological recordings - the extensive and operational observation of the snow cover and its properties, for which characteristic locations in the high Alpine terrain of the catchment were chosen. Therefore, the three AWS Bella Vista, Latschbloder, and Proviantdepot have been newly equipped with extensive automatic measurement systems to continuously observe various snow cover properties. These comprise observations of snow depth (HS), snow water equivalent (SWE), layered snow temperature profiles, snow surface temperature, liquid and solid water content of the snowpack, as well as snow drift (see Tabs. 1, 2, and 3). The data at all three automatic weather and snow stations (AWSS) is recorded in 10 min intervals and transmitted by means of mobile network GSM (Global System for Mobile Communications). In Tabs. 1, 2, and 3 the technical sensor specifications are listed in detail. The data of the AWSSs are visualized in real-time at https://avalanche.report/weather/measurements (last access: June 13, 2024) and https://lawis.at/station/ (last access: June 13, 2024).

### 4.1 Bella Vista

The Bella Vista AWSS (2805 m a.s.l., 46.78284°N, 10.79138°E, Tab. 1) is located in close vicinity to the "Schöne Aussicht Schutzhütte" (Fig. 1). It is located exactly at the central ridge of the Central Eastern Alps, the weather divide between the Northern and Southern Eastern Alps. Generally, the region belongs to the rather dry inner Alpine climate zone. The station is built in small scale heterogeneous terrain at a gentle slope in a barren, rocky landscape. It is affected by wind and frontal systems from both, northern and southern directions. In September 2017, the sensors for air temperature, relative humidity, wind speed, and air pressure were replaced by new instruments (Tab. 1). The Vaisala WXT520 instrument was replaced by E+E EE08 sensors for air temperature and relative humidity, by a Kroneis 262 instrument for wind speed and direction, and by a Young 61302V sensor for atmospheric pressure. The AWSS has been extended by additional snow measurement systems in October 2019 and September 2020. A snow scale to measure SWE, an ultrasonic HS sensor, and a snow temperature profiler have been installed in a depressed location near the main station that is prone to snow accumulation by wind-drift (Fig. 2, top). The data recorded by the new instruments complement the existing measurements of SWE (by means of a snow pillow), HS (by means of an ultrasonic ranger) and snow temperature profile (by means of a series of temperature sensors at the base level and in 20, 40, 60, and 100 cm from the ground) located at the main station. The location of the main station is rather exposed and therefore prone to snow erosion by wind. The relation between the exposed and sheltered snow measurements allows for an assessment of the timing and amount of wind-driven snow redistribution. This technique is illustrated by the data analysis

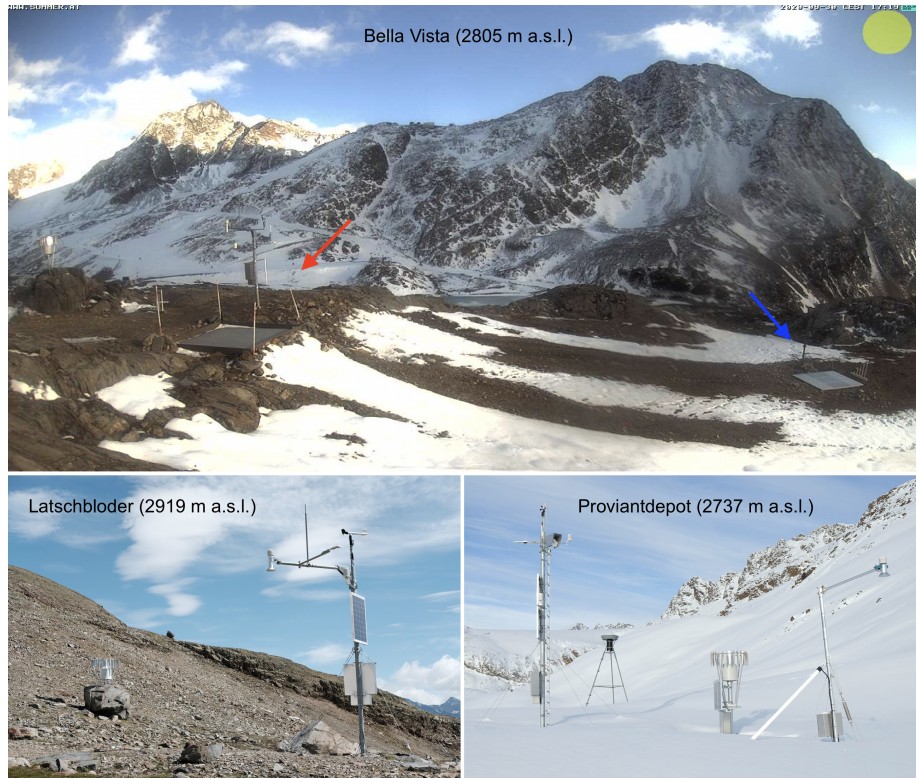

**Figure 2.** Top: webcam image from the Bella Vista station (2805 m a.s.l.). The red arrow marks the main "exposed" AWSS. The blue arrow marks the additional snow measurements (HS, SWE, and snow temperatures) in the slight depression ("sheltered" location). Bottom left: the Latschbloder AWSS (2919 m a.s.l.) with an Ott Pluviometer on the left. Bottom right: the Proviantdepot AWSS (2737 m a.s.l.). The ultrasonic snow depth sensor on the right instrument is part of the Snow Pack Analyzer (SPA). The snow scale is buried right in front of the photographer besides the SPA. There is a second snow depth sensor at the main mast (not visible from this angle). Behind the main mast the old totalizing rain gauge can be seen, and in the background the Kesselwandferner (left, behind the main mast).

of a blowing snow event in December 2020 (Sect. 7.4). In addition, a new instrument to directly measure the particle flux of drifting snow by means of an acoustic sensor (SND - Snow Drift Sensor) has been installed at the main station in September 2020. Fig. 3 (top) shows the installed sensors and the concept of the exposed and sheltered locations in a schematic overview.

## 4.2 Latschbloder

The Latschbloder AWSS (2919 m a.s.l., 46.80106°N, 10.80659°E, Tab. 2, Fig. 1) is located on a gently sloped plateau below the "Rofenbergköpfe" (3229 m a.s.l.) and was chosen as a meteorologically representative measurement site for the regional climate that is not largely influenced by steep surrounding slopes and the corresponding local wind systems (Fig. 2, bottom left). It is located near a totalizing rain gauge that was installed in 1965 and has been equipped with an ultrasonic snow depth sensor (USH-8) in 2017. In September 2017, the sensors for air temperature, relative humidity, wind speed, and air pressure at

Latschbloder were replaced by new instruments (Tab. 2). The Vaisala WXT520 instrument was replaced by E+E EE08 sensors for air temperature and relative humidity, by a Young 05103 instrument for wind speed and direction, and by a Young 61302V sensor for atmospheric pressure. In September 2020 an automatic snow temperature profiler that records temperature in the snow cover at the base level and in 20, 40, 60, and 100 cm from the ground has been installed. The current ensemble of sensors is illustrated in Fig. 3.

### 4.3 Proviantdepot

The Proviantdepot AWSS (2737 m a.ssl., 46.82951°N, 10.82407°E, Tab. 3, Fig. 1) is located on a flat section of a south-facing slope underneath the "Mittlere Guslarspitze" (3126 m a.s.l.) halfway between the summit and the "Hochjoch-Hospiz" (2413 m a.s.l.). The ensemble of instruments is situated on a flat plane built by an old moraine at the left orographic side of the Kesselwandferner. The slope behind and below the instruments faces south, and from the location of the station one has a panoramic view of the total area of the Hintereisferner. The new instruments are located approximately 20 m east of the totalizing rain gauge that has been in operation since 1952 (Fig. 2, bottom right). The station comprises several operational snow cover sensors (Fig. 3). HS is measured by a USH-9 ultrasonic device, SWE by means of a SSG-2 snow scale. The temperature of the snow surface is continuously measured by an infrared sensor (SIR). The layered snow temperatures are recorded analog to the other stations by a SCA temperature profiler at the base level and in 20, 40, 60, and 100 cm from the ground. A SPA-2 Snow Pack Analyzer records volumetric contents of solid and liquid water of the snow cover based on measuring the dielectric constants at different frequencies along a flat strap sensor that is spanned within the snowpack. At the Proviantdepot station, the SPA is installed in a configuration with one flat band spanned diagonally to measure bulk snowpack properties (see Fig. 2, bottom right; data series "S1") and one horizontally 10 cm above the ground to measure base layer properties (buried in Fig. 2, bottom right; data series "S2").

## 5 Data processing, coverage, and uncertainties

### 5.1 Data processing

The time series (September 2013 to August 2023) recorded at the AWSSs described in Sect. 4 are available as 10 min raw data under the Creative Commons Attribution License CC BY 4.0 at a GFZ Data Services repository: https://doi.org/10.5880/fidgeo.2023.037. They include data from 2013 to 2017 that has previously been published in a PANGAEA repository (https://doi.org/10.1594/PANGAEA.876120) and extend it by the new recordings from 2017 to 2023. Future data will be uploaded annually to the GFZ Data Services repository. Time zone for all data is UTC+1.

The data stream from the AWSSs to the repository is described in the following. The signal retrieved by the sensors is translated and stored on the data logger in 10 min temporal resolution. The 10 min data are the result of high frequency measurements of the prior 10 minutes that are processed in the logger. Depending on the measured variable, average (e.g., temperatures, relative humidity, snow depth, SWE, radiation), total (precipitation) or maximum (wind gust) are processed and

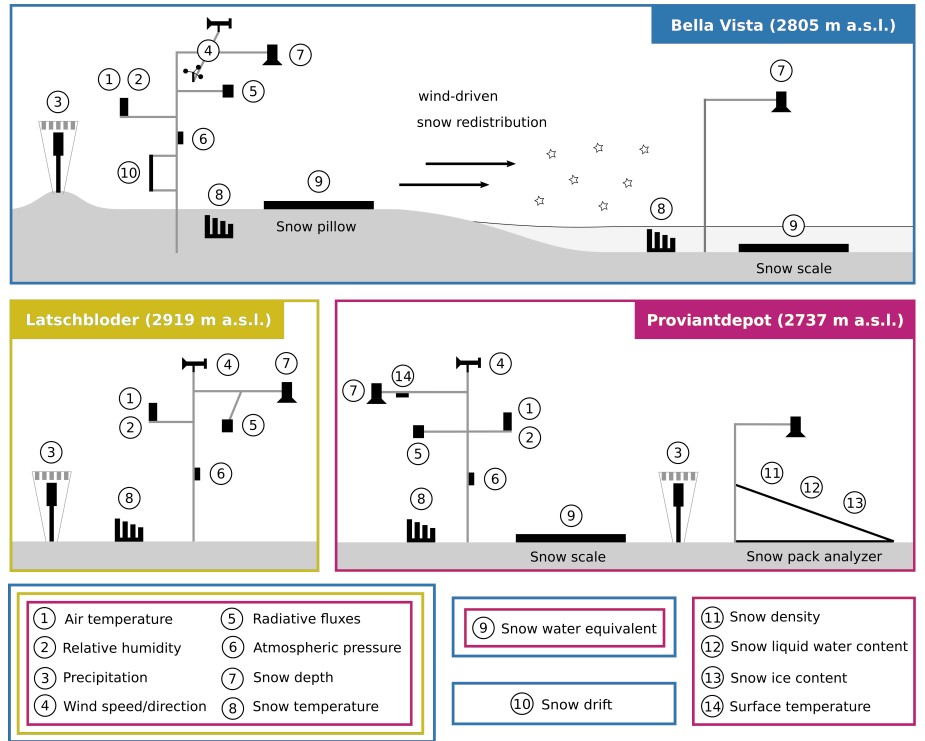

**Figure 3.** Schematic overview of the three AWSS Bella Vista, Latschbloder, Proviantdepot. The colour-coded boxes around the numbered variables show the respective equipment installed at each station. The relative arrangement between the instruments in the scheme does not correspond exactly to reality for display reasons.

stored. The high frequency of measurement is 1 minute, except for wind speed and direction, where it is 1 second. The 10 min data is continuously transferred by GSM to a web server. Several basic processing steps are applied to the raw data. These are correction of long-wave radiation for sensor temperature, decumulating the precipitation measurements, a thorough check for obvious measurement errors and a semi-automatic correction for erroneous values. This is done by applying thresholds for unrealistic values or unrealistically large leaps between consecutive time steps, as well as for periods of constant values (e.g. for wind speed and direction in case of frozen sensor). No gap filling methods have been applied for the existing periods of missing values in the data time series.

## 5.2 Temporal coverage

Fig. 4 shows the availability of data for the three stations from September 2013 to August 2023 after removing erroneous values. The bars indicate measured variables, the date of installation for the respective sensor and existing data gaps. The Latschbloder AWSS was installed in September 2013, followed by Bella Vista in July 2015 and finally Proviantdepot in October 2019. The snow specific measurements were implemented in September 2017 at Bella Vista and Latschbloder, the specific snow drift

installations at Bella Vista in summer 2020. The existing data gaps are mainly due to intermittent logger failures. The frequent short gaps in atmospheric pressure readings at the Bella Vista and Latschbloder site are caused by frequent single 10 min data points missing (unidentified malfunction of the logger). The large missing data periods at the Bella Vista station were caused by recurring lightning damage to the logger and the pluviometer.

## 5.3 Data uncertainties

Uncertainties in the data from the automatic stations origin from different sources which are unraveled to a certain degree by considering the following distinction. The data quality of all variables depends on the accuracy of the applied sensors. For most of them, these accuracies were tested by the respective manufacturer under laboratory conditions and they are listed in Tabs. 1, 2, and 3 for all three stations and variables. The second and larger source of uncertainty is introduced by measurement errors depending on I) the measurement principle of the respective sensor, II) by the characteristics of the variable to be recorded in the given high Alpine environment and climate conditions, and III) by unforeseen difficulties and effects caused by the combination of the former two. The latter source of uncertainty is a wide topic in hydrometeorological research and will not be tackled in this publication. A full quantitative assessment of these uncertainties for the presented data is for the most part not possible in the given setting. However, we suggest a qualitative categorization in three qualitative uncertainty classes (small, medium, high) depending on the observed variable and sensor:

– Small uncertainty: if effects like freezing of the sensor or coverage by snow are detected and masked (done in the presented data set), the following variables can be considered to show comparably small uncertainties which are close to the manufacturer accuracies: air temperature, relative humidity, radiative fluxes, wind speed, gusts and direction, atmospheric pressure, snow depth, snow temperature profile, and surface temperature.

– Medium uncertainty: Precipitation and snow water equivalent can be considered as data encompassing medium uncertainty, as there are sources of errors such as precipitation undercatch or snow bridging that are known but are difficult to identify and to correct for (not done in the presented data set).

– High uncertainty: data from the experimental snow sensors include high uncertainties and potentially show very large errors: snow drift (SND), snow density (SPA), and snow liquid water and ice content (SPA). The quality and potential of the data from the experimental sensors is discussed in the detail in the following respective sections.

## 6 Meteorological data

Fig. 5 presents the main meteorological variables measured at the three stations Bella Vista, Latschbloder, and Proviantdepot from September 2017 to August 2023. The data prior to September 2017 has been described by Strasser et al. (2018). All data shown here was recorded by the most recently installed sensors (see Tabs. 1, 2, and 3). Daily averages are shown for air temperature, relative humidity, short-wave radiation, long-wave radiation, and wind speed, as well as monthly totals for

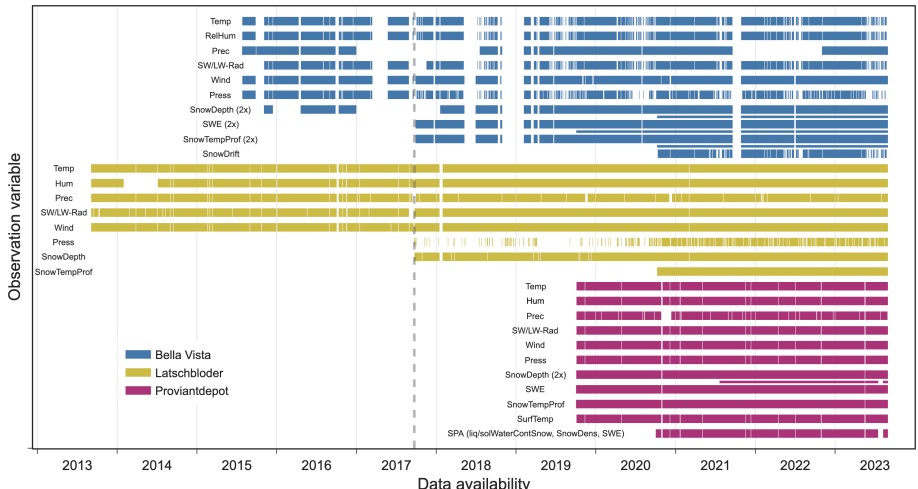

**Figure 4.** Data availability for the three AWSS Bella Vista, Latschbloder, and Proviantdepot for all measured variables 2013 to 2023. Some variables are recorded with two different sensors at the same location. These are indicated by (2x) and the narrow bars show the data availability for the second sensor. The vertical dashed line highlights the beginning of the data period presented here (September 2017). The plot is based on daily aggregations of the data. If one 10 min value is missing on a specific day, the entire day is classified as missing data in this figure. The frequent short gaps in atmospheric pressure readings at the Bella Vista and Latschbloder site are caused by frequent single 10 min data points missing (unidentified malfunction of the logger). The large data gaps at the Bella Vista station were caused by recurring lightning damage.

precipitation. Temperature and short-wave incoming radiation show a typical seasonal cycle. Temperature values range from
below -20 °C during winter to approximately +10 °C in the summer seasons. Absolute minimum and maximum values in the
10 min data were recorded both at the Bella Vista station with -30.4 °C (February 27, 2018, 3:40 am) and +18.8 °C (July 24,
2019, 4:20 pm), respectively. The same maximum temperature of +18.8 °C was observed again at the Proviantdepot station
at August 24, 2023, 3:00 pm. Maximum wind speeds were measured during a storm in October 2018 at the Latschbloder
station with mean wind speeds of 21 m s$^{-1}$ in the 10 min data (18 m s$^{-1}$ hourly, 9 m s$^{-1}$ daily), and wind gusts reaching 45 m s$^{-1}$.
Wind speeds exhibit a weakly pronounced seasonal pattern with higher values occurring in the winter months. This reflects
the general regional situation of a higher storm frequency in winter caused by frequent distinct low-pressure systems and
Foehn events. Short-wave outgoing radiation (Fig. 5) is depending on incoming radiation and strongly controlled by snow
coverage on the ground. Values from 200 W m$^{-2}$ reflected (outgoing) radiation in spring suddenly drop to values as low as
10 W m$^{-2}$ while incoming radiation is between 200 and 300 W m$^{-2}$. This occurs in periods when the ground becomes free of
snow in spring (May/June) and albedo instantly decreases. The largest precipitation amounts of around 150 to 200 mm/month
typically fall during the summer months (June, July, August), whereas during the remaining year, average monthly precipitation
totals are approximately 50 mm. For the year 2020, annual precipitation totals are 827 mm at the Bella Vista, 960 mm at
the Latschbloder, and 864 mm at the Proviantdepot station. In the years 2020 to 2022, the values range between 800 and

900 mm annual precipitation for all three stations. In the preceding years (2013 to 2019), there are several values of around 1300 mm annual precipitation at the Bella Vista and Latschbloder station, and 1590 mm in 2014 at the Latschbloder station. The two totalizing rain gauges that are located close to the stations recorded long-term annual precipitation totals of 1012 mm (Latschbloder, 1965 to 2016), and 941 mm (Proviantdepot, 1952 to 2016), respectively. The precipitation values are uncorrected and might - despite the use of heated Pluviometers (Bella Vista and Proviantdepot) equipped with wind shelters (all three) - suffer from precipitation gauge undercatch that is typical for high mountain observations with high wind speeds and a large amount of solid precipitation. When comparing winter precipitation to measured maximum SWE values (not shown here), a strong interannual variability in the estimated undercatch amounts can be identified depending on the meteorological characteristics of the respective winter season.

## 7  Snow cover data

In the following section, we present the newly established data sets obtained by the various snow observation sensors from September 2017 to August 2023. Whereas the observations of snow depth, snow water equivalent, and snow temperature have proven their reliability and are operationally used by the avalanche warning and hydrographic services of Tyrol (Austria) and South Tyrol (Italy), the accuracy of the snow drift sensor as well as of the liquid and solid water content measured by the Snow Pack Analyzer are still under evaluation.

### 7.1  Snow depth and water equivalent

All available HS and SWE data for the three stations is shown in Fig. 6 (September 2017 to August 2023). In Fig. 7 a) daily averaged snow depth values for the three stations Bella Vista, Latschbloder, and Proviantdepot are compared for the period September 2017 to August 2023. Generally winter HS varies between 0.5 and 2 m depending on station and year. At the Bella Vista exposed site usually less HS is measured than at the Latschbloder station. The Bella Vista sheltered site data (available since autumn 2020), shows significantly higher HS values than the exposed site. Maximum HS of 2.2 m were measured at the Bella Vista sheltered site in May 2021. HS of close to 2 m were measured at Latschbloder in May 2019, the maximum at the Proviandepot was observed in winter 2019/2020 in December 2019 and March 2020. The winter 2019/2020 shows large differences in maximum HS between Latschbloder (50 cm) and Proviantdepot (160 cm). These large variations can not be explained by measured precipitation data or meteorological conditions. It might be caused by differences in local terrain characteristics and wind effects: in a less pronounced extent the general pattern occurs again in the following winter seasons. Starting with the winter season 2021/2022, at the Proviantdepot station the HS measurements from the SPA instrument are logged separately (referred to as HS SPA in the following). Despite being only few meters apart, significant differences in HS are observed from January to May 2022 between the ultrasonic HS sensor USH-9 at the main mast and the HS sensor of the SPA. The USH-9 at the main mast is located approximately 4 m beside the snow scale and the SPA (Fig. 2, bottom right). By means of webcam images in the melt out periods of 2020 and 2021 (not shown here) we identified that there were snow patches covering the scale and the SPA while the surroundings were already free of snow and measured HS by the USH-9 at the main

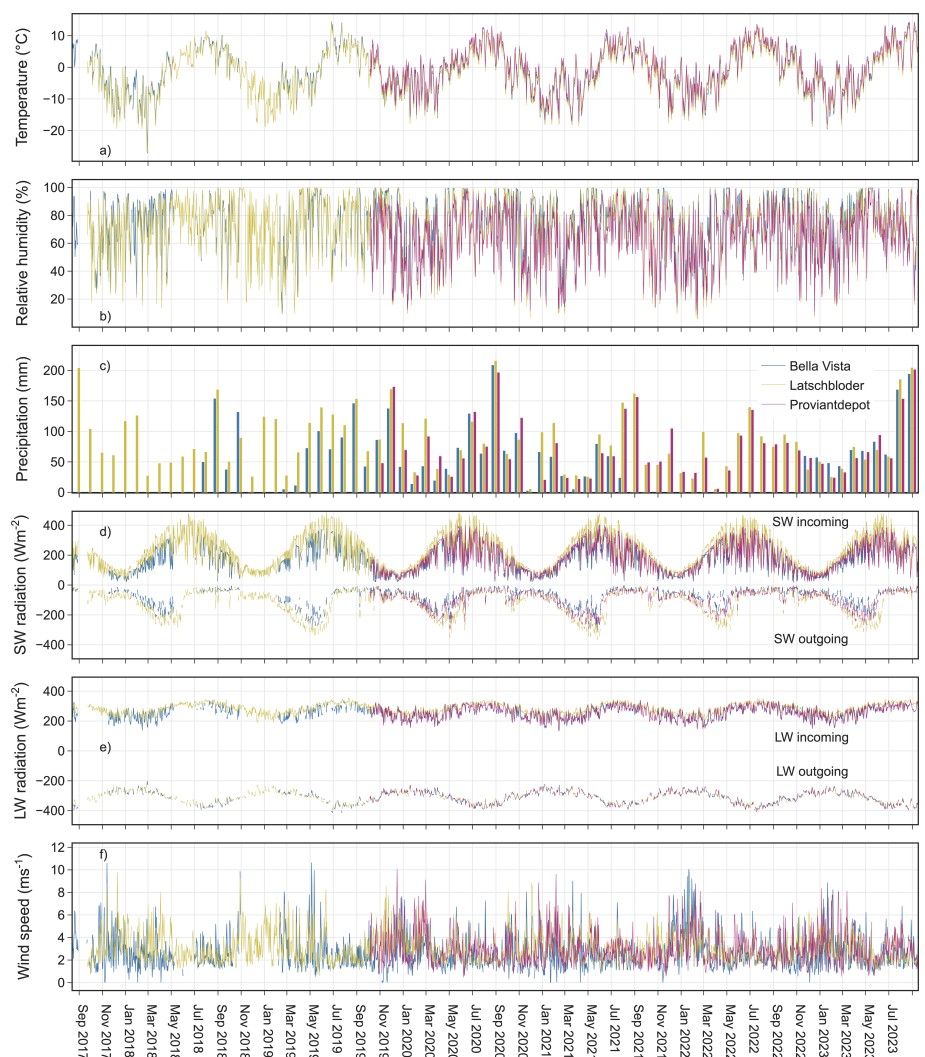

**Figure 5.** Main meteorological variables (daily averages) at the three stations Bella Vista, Latschbloder, and Proviantdepot (September 2017 to August 2023). Air temperature (a), relative humidity (b), precipitation (c) (monthly totals), short-wave radiation (d), long-wave radiation (e), and wind speed (f).

mast was zero. This was caused by very small-scale wind-driven snow redistribution. Despite the differences in HS between the two nearby measurements in the season 2021/2022, the melt out dates are very close (4 days later at the SPA). In the winter 2022/2023, no such differences caused by wind-driven snow redistribution occur.

Fig. 7 b) shows daily averaged snow water equivalent measurements. The large difference between the Bella Vista exposed
and sheltered sites is clearly visible. It is noted that SWE at the Bella Vista is measured by two different instruments, a snow pillow at the exposed and snow scale at the sheltered site. This might introduce additional deviations but to a much smaller extent than the recorded differences. The Bella Vista sheltered site shows comparable amounts of SWE to the Proviantdepot

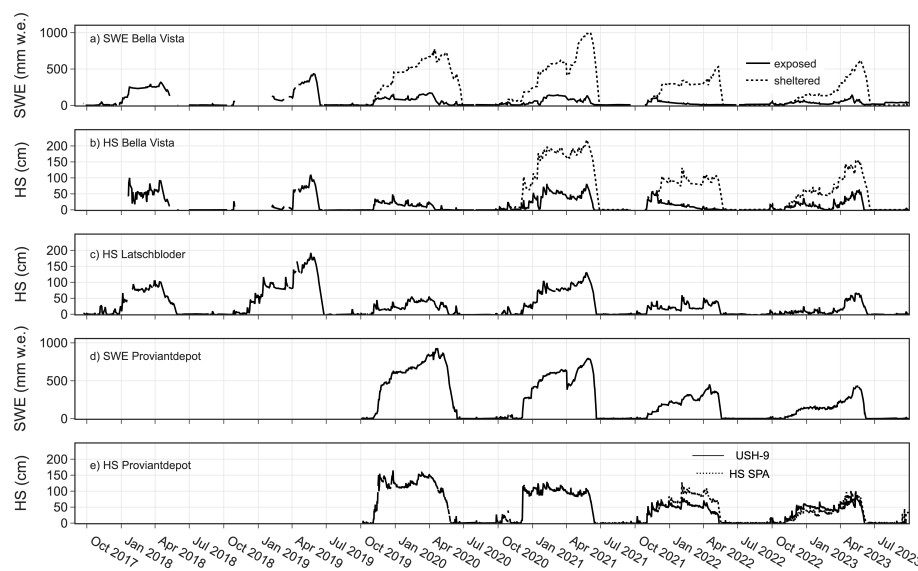

**Figure 6.** Measured SWE and HS (where available) at the Bella Vista double snow station setup (a and b), at the Latschbloder (c) and Proviantdepot (d and e) stations from September 2017 to August 2023.

station and very similar accumulation and ablation dynamics. However, the date of melt out in spring occurs consistently later at the Bella Vista sheltered site than at Proviantdepot. When the snow becomes isothermal from the second half of March

on, some erroneous values due to building and weakening of bridging snow structures can be found. A very distinct event of such bridging is discernible at the beginning of April 2021 at the Bella Vista sheltered location and at the Proviantdepot site where measured SWE sharply drops. The meteorological measurements in Fig. 5 reveal large air temperature fluctuations in that period which generally fosters bridge building. These processes typically occur during the transition from a cold to a warm snow cover when the isothermal front reaches the snow-ground interface (Johnson et al., 2002, 2004). A method to correct

these errors based on their detection using change rates of HS, SWE, and snow density is suggested by Johnson et al. (2004). As these cases are not always unambiguous, they are not corrected here.

     The locations of all three stations have been chosen to be representative for their surroundings and for the regional climate and snow conditions at the respective elevation. Due to the heterogeneity of the terrain, large differences in snow cover occur at a very small spatial scale. These small scale differences are captured by the double station setup at the Bella Vista site. The

presented data reveals that the location of the Latschbloder snow measurements can be prone to wind-driven erosion of snow. For most winter seasons, HS at Latschbloder is close to the Bella Vista exposed site. In contrast, the Proviantdepot station shows a more sheltered characteristic and the measurements are closer to the Bella Vista sheltered site. However, there are single winter seasons that do not show these effects. This emphasizes the strong interannual variability in the spatio-temporal variations of snow cover dynamics in high Alpine terrain.

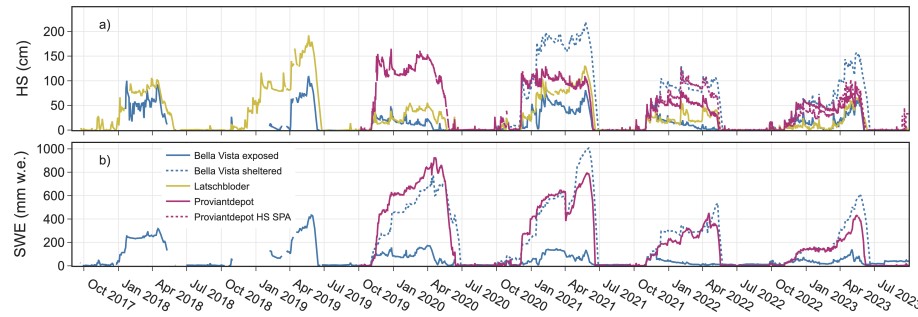

**Figure 7.** Daily averaged HS (a) and SWE (b) measurements (September 2017 to August 2023) at the three stations Bella Vista, Latschbloder and Proviantdepot.

## 7.2 Continuous snow temperature profile

A snow temperature profile recorded by the SCA temperature profiler for the whole snow covered period in winter 2019/2020 at the station Proviantdepot is shown in Fig. 8 Ib). Heights above the ground are 0, 20, 40, 60, 80 and 100 cm, respectively. The corresponding snow depth (Fig. 8 Ia) shows a typical course over the season. The first large snowfall events in the beginning of November result in a snow depth of 40 cm that grows to 140 cm with more snowfall until the end of the month. In the following, the snow cover settles slowly and periodically increases again with single snow precipitation events. Snow depth reaches its seasonal peak of 160 cm at the end of December. A long period with only small sporadic snowfall amounts and constant snow settling follows during January and February, where HS varies between 110 and 130 cm. Another large snowfall at the end of February leads to a HS of 150 cm. After some settling and little new snow periods, the melting period starts at the beginning of April. Snow melts constantly - only interrupted by some late snowfall at the beginning of May - until HS is 0 cm at the end of May. The described behaviour and distinguishing melt and settling periods can well be reconstructed using the SWE measurements in Fig. 8 Ia). SWE steadily increases during snowfall events and stagnates in midwinter when snow is settling but not melting. SWE decreases with the start of the melting period in April. The difference of the points in time of SWE and HS reaching zero can be explained by the ultrasonic HS sensor being approximately 4 m beside the snow scale at the Proviantdepot station (Fig. 2, bottom right) and a heterogeneous melt out pattern. During most of the presented season SWE exhibits comparably large values which do not match measured HS. By means of webcam images (not shown here) we identified that there was a snow patch covering half of the scale while the surrounding was already free of snow. This phenomenon occurs again in the winter season 2020/2021 (Fig. 8 IIa) but not in the seasons 2021/2022 and 2022/2023 which both generally have less snow than the previous years (see Figs. 6, and 7). Snow temperature at the base (0 cm) is at 0 °C throughout the whole snow covered period (Fig. 8 Ib). The elevated temperature sensors at 20, 40, 60, 80, and 100 cm obviously show strong diurnal variations when they are not covered by snow, i.e. the air temperature is measured at the beginning and end of the snow season (not shown in Fig. 8). As soon as the sensors are covered by snow, the temperature signal is dampened. A clear snow temperature stratigraphy with warmer (negative) temperatures in deeper levels (closer to the ground) develops and is retained throughout the season. The layer closest to the surface (100 cm) is influenced the most by

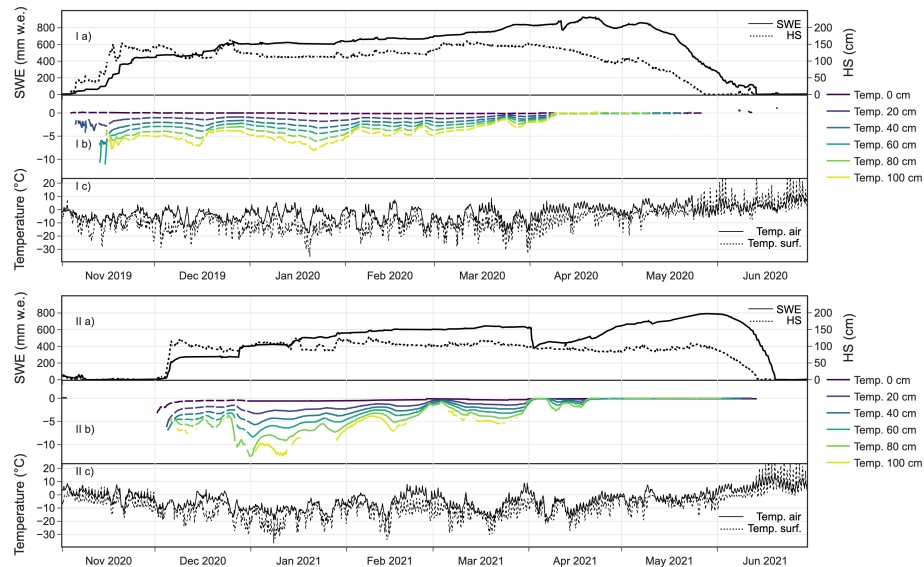

**Figure 8.** Hourly measurements of HS and SWE (a), snow temperatures at the base and in 5 height levels above the ground / in the snow cover (b), and snow surface and air temperature (c) at the Proviantdepot station from Nov 1, 2019 to Jun 30, 2020 (I) and from Nov 1, 2020 to Jun 30, 2021 (II). Snow temperatures are shown only if HS exceeds the respective sensor height. It is noted that during the presented seasons SWE exhibits comparably large values which do not match measured HS. SWE melt out date is two weeks later than HS in 2020 and five days later in 2021. This is caused by wind-blown snow deposited on the SWE scale.

prevailing air temperatures and cools down to -8.1 °C on January 21. This cold minimum in the layer approximately 20 cm below the snow surface follows air temperature with a time lag of 1 day. Very cold air temperatures of -17 °C were measured in the night preceding January 20. A minimum snow surface temperature of -36.6 °C was recorded after that cold night at 10:40 pm (Fig. 8 Ic). This time lag is carried on into the deeper snow layers. This dampening effect can be observed in the data in both directions, i.e. when the snowpack is cooling or warming. The data show a very sharp point in time when the snowpack becomes and stays isothermal, i.e. all layers are at 0 °C (April 9, 2020). This generally marks the beginning of the melting period and HS starts to decrease.

The seasonal pattern is repeated similarly in the following winter season 2020/2021 (Fig. 8 II). Snow coverage begins a month later at the beginning of December. A first significant warming of the snow pack occurs in the beginning of March where it is close to becoming isothermal before cold conditions cool the snowpack again (Fig. 8 IIb). At the end of March, a very warm period with air temperatures of up to almost 10 °C (Fig. 8 IIb) leads to a rapid warming of the snowpack. With colder air temperatures and an isothermal snowpack shortly after at the beginnig of April, bridge building effects lead to the measurement error (sharp decline of SWE) described in Sect. 7.1. Following this cold period, isothermal conditions develop in the second half of April and lead to the melt out in June.

## 7.3 Liquid and solid water content (Snow Pack Analyzer)

Fig. 9 shows measurements of the Snow Pack Analyzer (SPA) at the Proviantdepot station. The measurement principle of the instrument was first described by Stähli et al. (2004). Based on the different dielectric constants of ice, water, and air, measurements of impedance along flat ribbon sensors allow to infer snow density, as well as separate liquid and solid water content. These flat ribbon cables can be installed diagonally to measure bulk snowpack properties or horizontally for measuring in single snow layers. Stähli et al. (2004) showed robust agreement of measured snow density and liquid water content with manual snowpack observations. Using a simultaneous measurement of HS, SWE can be calculated. Egli et al. (2009) evaluated the instrument regarding its performance in measuring SWE compared to other automatic systems. They found that the SPA shows a good performance with respect to average SWE estimations with some interannual deviations depending on the snowpack formation and the placement of the flat band cable within. Schattan et al. (2017) compared SPA snow density to sporadic manual measurements and showed good results when HS is above a certain threshold (50 cm in their specific application).

At the Proviantdepot station, the SPA is installed in a configuration with one flat band spanned diagonally (data series "S1") and one horizontally 10 cm above the ground (data series "S2"). The data from the diagonally spanned sensor exhibits a lot of noise when HS is below a certain threshold. In the following analysis, the S1 time series is omitted when HS is below 30 cm and the S2 data when HS is below 20 cm (these thresholds were not applied to the raw data in the repository). Due to a logger failure, SPA data is not available before October 2020. Here we show the filtered data from October 2020 to August 2023. Fig. 9 a) shows solid and liquid volumetric water content measured by the SPA for the two flat band sensors S1 and S2. Ice content slowly increases with settling of new snow. Liquid water content increases during melting periods with maximum values of 5 to 10 vol %. The corresponding snow density measurements are shown in Fig. 9 b) and for the two winter seasons 2021/2022 and 2022/2023 compared to snow density that was independently calculated by measured SWE (snow scale) and HS at the SPA (not available for winter 2020/2021). S2 snow density at the base of the snow pack ranges from 100 kg m$^{-3}$ at the beginning of the season to 350 kg m$^{-3}$ in May. Density steadily increases with almost no fluctuations. Contrarily, the diagonal band data S1 shows strong fluctuations throughout the winter season because it is influenced by processes at and near the snow surface. The S1 density reaches a maximum value of 520 kg m$^{-3}$. In the winter months December to February, the calculated density is in good agreement with the SPA measurement with a slight offset to higher values (Fig. 9 b). However, it strongly deviates in the spring melting season (March to May) with the SPA measurement maxing out at 300 kg m$^{-3}$ in the two winter seasons 2021/2022 and 2022/2023. A possible reason is a comparably low HS during these winters which leads to an underestimation of density by the SPA, and at the same time to an overestimation of calculated density at the very end of the melting period. In the previous winter season 2019/2020 this effect does not occur in the data. The course of SWE measured by the SPA is in line with SWE recorded by the snow scale with slightly larger values at the scale (Fig. 9 b). It is noted that the snow scale measurements of SWE and snow density are bulk values for the whole snowpack independent of HS while the SPA measures properties only for the layer around the respective flat band, i.e., the base layer of the snow pack for S2, and the layer that is spanned by the diagonal band for S1, respectively.

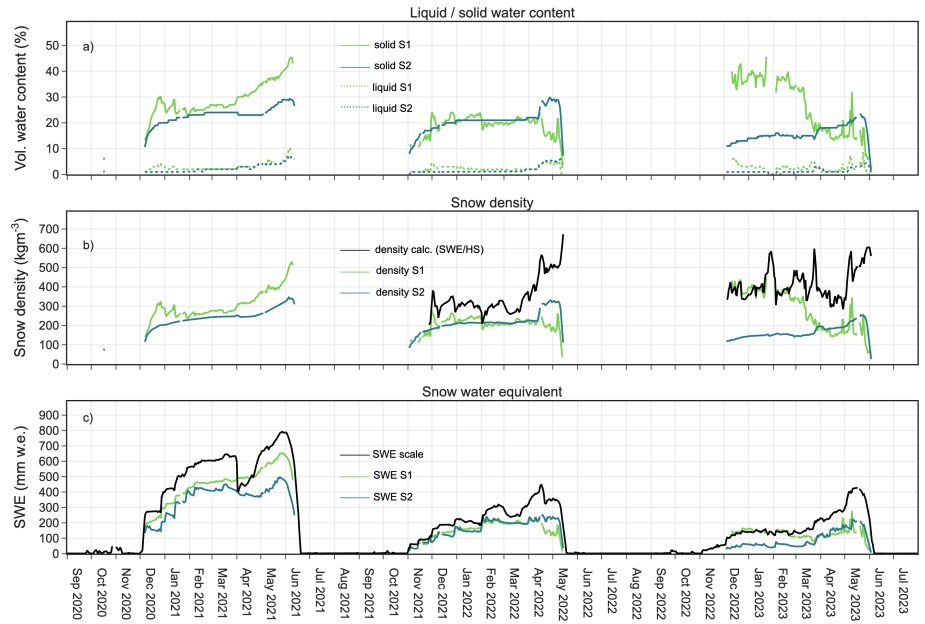

**Figure 9.** Liquid and solid water content measured by SPA (a), snow density as measured by SPA and calculated from independent measurements of SWE and HS (b), and SWE measured by SPA and by the snow scale (c). All data: hourly averages at the Proviantdepot station from Nov 25, 2020 to Dec 31, 2020. S1 is data from the diagonally spanned flat band, S2 from the flat band that is spanned horizontally 10 cm above the ground. S1 data is filtered for HS below 30 cm, S2 for HS below 20 cm. Calculated density in (b) is not available in winter 2020/2021 because HS data from the SPA was not logged before July 2021.

### 7.4 Snow drift measurements

In Fig. 11, a snow drift event is analysed based on the data recorded by the described instrumentation at the Bella Vista station. This includes data from an acoustic-based snow drift sensor (Fig. 10, SND). The system continuously records the snow particle flux in the air by registering the change in acoustic pressure caused by bypassing particles colliding with the cylindrical sensor tube. The measurement principle was first described by Chritin et al. (1999). The accuracy of different versions of the sensor in quantifying snow flux was discussed in literature by, e.g., Jaedicke (2001); Lehning et al. (2002); Cierco et al. (2007); Trouvilliez et al. (2015) with differing results. The study by Trouvilliez et al. (2015) evaluates the most recent generation of the sensor and shows large underestimation of measured fluxes during precipitation events. For periods without precipitation both, over- and underestimation of snow flux were observed, mostly depending on snow particle speed and densities. However, despite limitations in measuring absolute snow flux values, acoustic-based sensors are besides optical snow particle counters (Sato et al., 1993; Naaim-Bouvet et al., 2014) still the only way to continuously measure and detect drifting snow events with a certain reliability (He and Ohara, 2017). Specifically for event detection, acoustic snow drift sensors have proven to be reliable instruments that can withstand harsh conditions (Trouvilliez et al., 2015).

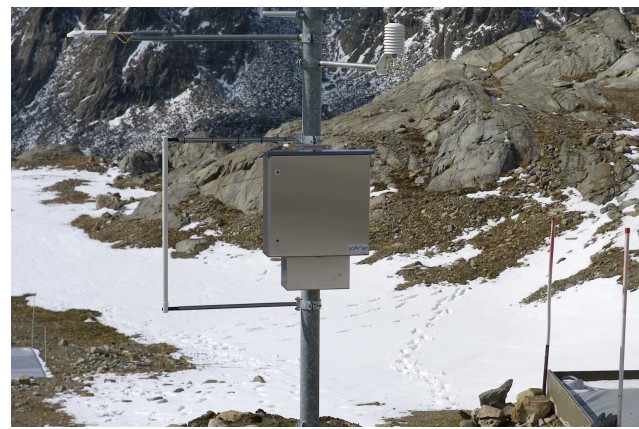

**Figure 10.** The SND snow drift sensor installation at the Bella Vista AWSS. The sensor elements are placed in the cylindrical tube that is vertically held in place by two mounting arms building a rectangular shape (left of the logger box). In the background (lower left in the photo), the snow scale in the slight depression is visible (sheltered site, sink for snow deposition). The snow pillow besides the main AWSS mast (lower right in the picture) measures SWE at the exposed location.

Cumulated measured snow flux per hour is shown in Fig. 11 d) during a blowing snow event at the Bella Vista station in the night of Dec 4, 2020. Fig. 11 also includes time series of hourly changes in SWE and HS, mean and maximum wind speeds, as well as precipitation and temperature from Dec 3 to 5, 2020. On Dec 5, 7 pm measured mean wind speeds increase from 2 to 8 m s$^{-1}$. Wind gusts range up to 22 m s$^{-1}$ during the storm that lasts the night and ends in the morning the next day (6 am). Temperatures are very low during the period varying from -7 to -12 °C. The storm is accompanied by snow precipitation in

the range of 0.5 to 1.7 mm/h. When comparing HS and SWE at the exposed and sheltered site, erosion and deposition of snow can clearly be identified in Fig. 11 a) and b). SWE at the exposed location decreases from Dec 4, 7 pm to 11 pm despite incoming snow fall. The time frame exactly matches the high wind speed recordings. Contrarily, SWE at the sheltered locations continuously increases as do the corresponding HS values. In contrast, HS at the exposed site increases with snowfall while wind speed is still low and strongly decreases with high wind-speeds shortly after (Dec 4, 7 pm) from 20 to almost 0 cm. In the

same time frame, HS at the sheltered site increases from 20 to 40 cm. These changes are illustrated by webcam images before and after the blowing snow event in Fig. 11.

     The snow drift sensor detects particle mass fluxes beginning on Dec 4, 4 pm and shows a strong signal when wind speeds are high. The lower rates until 7 pm suggest that the measured flux is produced by blowing snowfall that has not yet reached the ground as wind speeds are still low, e.g gusts of 5 m s$^{-1}$ at 6 pm. This is confirmed as both HS values stay constant or

increase in that period. HS at the exposed site decreases strongly from 7 pm to 11 pm. These are also the times with highest wind speeds and largest measured drift by the SND, i.e. in addition to blowing snowfall wind speed exceeds the snow erosion threshold. To analyse the snow mass flux in the air and the storage changes at the two ground measurements, we calculated the changes in SWE, as well as cumulated precipitation and snow particle drift flux from Dec 4, 7 pm to Dec 5, 1 am, i.e. the period with the strongest signals. The SND measured a snow particle flux of 22.8 kg m$^{-2}$ during that period. This corresponds

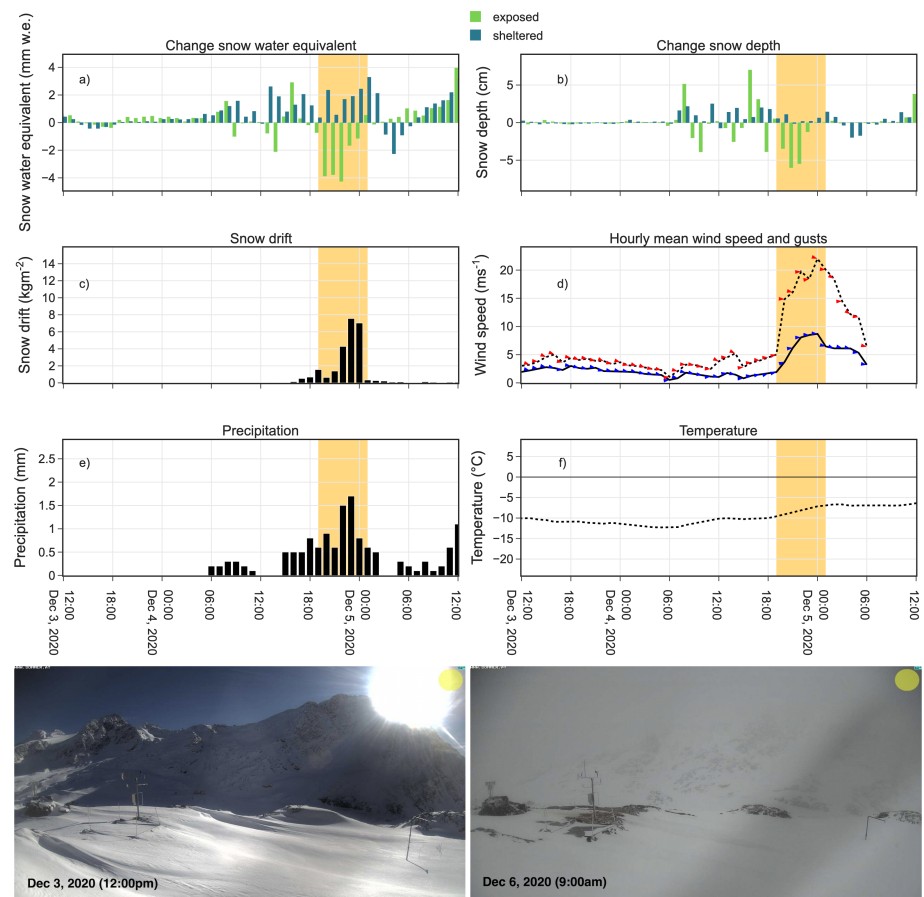

**Figure 11.** Measurement of a snow drift event at the Bella Vista station from Dec 3, 2020 (12:00 pm) to Dec 5, 2020 (12:00 pm): hourly data of changes in SWE (a) and HS (b) each at the exposed and sheltered location, snow drift (c), wind speed and gusts (d), precipitation (e), and air temperature (f). Wind directions for mean wind and gusts are indicated by blue and red arrows, respectively in (d). The period of the main blowing snow event is highlighted in yellow. The webcam images are taken before (left, Dec 3, 12:00 pm) and shortly after a blowing snow event (right, Dec 6, 2020, 9:00 am). The main AWSS (exposed location) can be identified in the left side of the pictures, the complementing, second SWE and HS measurements are located at the mast on the right side in the pictures (sheltered location).

well to the measured SWE changes of -14.1 mm w.e. / +12.2 mm w.e. at the exposed / sheltered point during the same time, particularly when adding the measured 5.9 mm of snowfall that were potentially hitting the instrument before reaching the ground. However, any snow drift values are measurements of a horizontal flux through an area in the atmosphere of 1 m² which can not be directly compared to snow depth or SWE changes per m² at point A or B on the ground. Measured wind direction shows west winds during the event. The exposed measurement site is situated northeast of the sheltered site, so only northeast winds would allow for a direct recording of snow erosion and deposition. As the sheltered site is a slight depression surrounded by more exposed areas, the recordings should be representative independent of wind direction. Nevertheless, for a full mass

balance of such a complex process, many more variables have to be taken into account (e.g., shear stress and strength, erosion threshold, suspension time, travel speed and distance, erosion and deposition zones). However, the instrument is able to detect the snow drift event that can also be observed by HS and SWE measurements and gives an estimate about the general extent of the event.

A five day period of blowing snow events in December 2021 is shown in Fig. 12. Wind direction is mainly east to southeast. From Jan 14, 8 am to Jan 14, 7 pm cumulated snow particle flux is measured as 33.3 kg m$^{-2}$. Snowfall is occuring mainly before that blowing snow event. The measured SWE changes of -79.9 mm w.e. / +115.9 mm w.e. at the exposed / sheltered point during the same period indicate an underestimation of measured particle flux. Three days later, from Jan 17, 4 pm to Jan 18, 12 am, a particle flux of 16.6 kg m$^{-2}$ was recorded, while SWE at both sites is decreasing (exposed: -12.7 mm w.e. / sheltered: -21.0 mm w.e.). A third event of snow redistribution is discernible on Jan 16 around noon (not marked in Fig. 12). The increase in SWE without precipitation suggests deposition of wind-driven snow on the scale of the sheltered site. The snow drift sensor shows only a very small signal and does not capture the redistributed snow amounts. These differing results exemplarily display the inconsistencies in the measurements of the SND and generally underline the challenges in measuring the complex process of wind-driven snow redistribution. Therefore, we will further investigate the potential of an automatic detection and assessment of magnitude of blowing snow events and a possible statistical relation to describe changes in HS / SWE at the two measurement locations in dependence of measured snow drift values. These results obviously are neither a full assessment of the sensor accuracy, nor an exact measurement of transported snow mass from point A to point B, but rather a showcase of the potential of the sensor and the double station setup to detect (avalanche / snow slab-critical) snow drift events, and to estimate the amounts of wind-driven snow redistribution in the surrounding terrain. Lehning et al. (2002) pointed out the potential of this semi-quantitative approach for avalanche warning applications. The accuracy of the sensor in quantifying snow transport rates and its relation to redistributed snow on the ground and at the HS and SWE measurement points will further be investigated. Additional field campaigns will be carried out using a mobile terrestrial laser scanning device to measure the spatial distribution of snow depth in the small-scale heterogeneous terrain around the AWSS following the approach performed by Lundquist et al. (2024). They combined measurements of two acoustic snow drift sensors with repetitive terrestrial laser scans to record a complete picture of the redistribution processes. However, observing and quantifying snow erosion and deposition processes including closure of the mass balance for a certain domain of investigation remains a challenging task.

## 8   Conclusions

We presented data from three AWSS in the Rofental research catchment that comprise continuous meteorological and snow cover measurements in a high temporal frequency (10 min) over six winter seasons (September 2017 to August 2023). Data from the most recently installed station is available for four winter seasons (October 2019 to August 2023) and data from individual additional snow sensors for three seasons (September 2020 to August 2023). First and foremost, the extensive observational data set is unique in its composition and extent. The data has the potential to be used in different scientific fields, as well as in operational applications. The records of the standard meteorological variables combined with the various snow

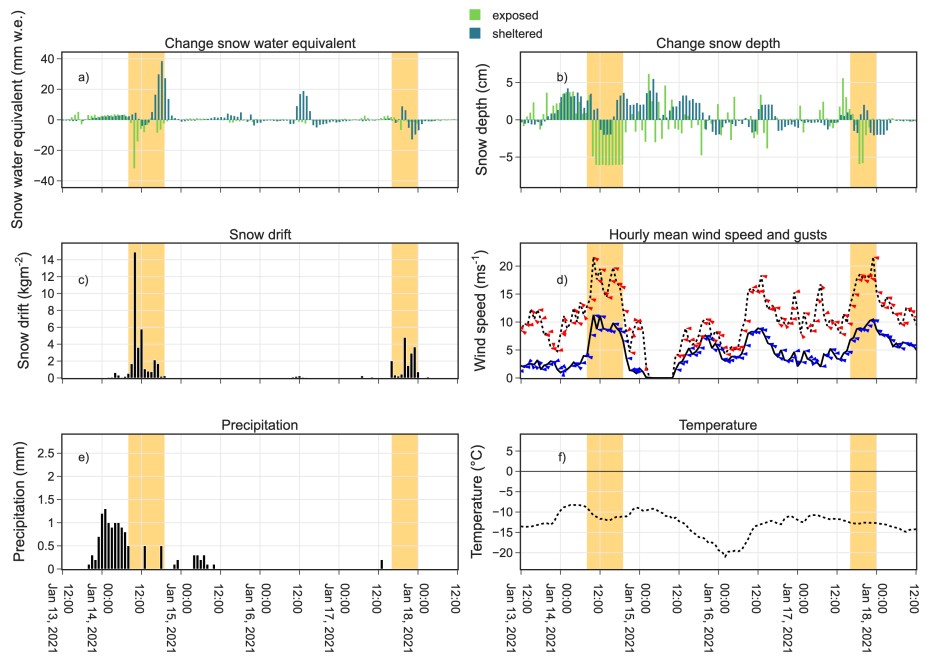

**Figure 12.** Measurements from a period with two snow drift events at the Bella Vista station. Data from Dec 13, 2021 (12:00 pm) to Dec 18, 2021 (12:00 pm): hourly data of changes in SWE (a) and HS (b) each at the exposed and sheltered location, snow drift (c), wind speed and gusts (d), precipitation (e), and air temperature (f). Wind directions for mean wind and gusts are indicated by blue and red arrows, respectively in (d). The periods of the blowing snow events are highlighted in yellow.

measurements at different high Alpine locations in addition to the rich other glaciological, hydrological, and meteorological data documented in Strasser et al. (2018) create a seldomly available data set. The presented data can be of use for tackling various research questions in the context of coupled climate and glacier evolution, snow and glacier hydrology, water resources in mountainous regions, or snow hydrological model development and evaluation. The continuous, automatic meteorological and snow observations are already used operationally for assessing regional avalanche danger and forecasting potential flood

events. Secondly, we presented insights into fairly new sensor technologies and an innovative setup of proven instruments to quantify snow redistribution. The data might be used, e.g., for automatic operational detection of avalanche-critical blowing snow situations to support avalanche warning services. Finally, the recordings of the snow sensors were used to draw pictures of several important seasonal snow processes in the region at the event-scale. All meteorological and snow monitoring installations in the Rofental are intended to stay in operation for the long-term. We will continue to provide the recordings to the community

via the GFZ repository.

## 9 Data availability

The data set presented here is available under the CC BY 4.0 license at GFZ Data Services: https://doi.org/10.5880/fidgeo.2023.037 (Department of Geography, University of Innsbruck, 2024). It extends a previous data collection from the catchment which is hosted under the same license at PANGAEA: https://doi.org/10.1594/PANGAEA.876120 (Strasser et al., 2017).

*Author contributions.* US, TM, and MW designed the station network and sensor concepts and conducted the installations. ER contributed in station installation and data analyses and visualization. MW compiled and analysed the data, and prepared the manuscript with contributions from the three co-authors.

*Competing interests.* The authors declare that no competing interests are present.

*Acknowledgements.* The University of Innsbruck as well as many colleagues and students supported the installation of the stations and
460 instruments as well as their maintenance. In particular, we thank Rainer Prinz for support in field work, Carsten Becker for support in data processing, and Paul Grüner and his team from the "Schöne Aussicht Schutzhütte" for providing comfort and very valuable support during the station installation and maintenance work. The LTSER platform Tyrolean Alps belongs to the national and international long term ecological research network (LTER-Austria, LTER Europe and ILTER). The infrastructure is financially supported by the University of Innsbruck, Faculty of Geo- and Atmospheric Sciences and is part of its Research Area "Mountain Regions". The publication of this paper is
465 supported by the University of Innsbruck.

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

**Table 1.** Climate and snow variables recorded by the sensors installed at the station Bella Vista (2805 m a.s.l., 46.78284° N, 10.79138° E). Accuracy according to technical data sheets of the manufacturers. Temporal resolution of the data records is 10 min. The station has power supply from the nearby "Schöne Aussicht Schutzhütte".

| Variable | Sensor | Period of operation | Measurement interval | Resolution and accuracy | Unit |
|---|---|---|---|---|---|
| Air temperature | E+E EE08 (ventilated) | Since Jul 2015 | 1 min 10 $\text{min}^{-1}$ (avg) | < 0.5 °C[2] | °C |
| | Vaisala WXT520[1] | Jul 2015 to Sep 2017 | 1 min 10 $\text{min}^{-1}$ (avg) | 0.1 °C $\pm$ 0.3 °C | |
| Relative humidity | E+E EE08 | Since Jul 2015 | 1 min 10 $\text{min}^{-1}$ (avg) | $\pm$ 2 % RH (0–90 % RH), $\pm$ 3 % RH (90–100 % RH) | % |
| | Vaisala WXT520[1] | Jul 2015 to Sep 2017 | 1 min 10 $\text{min}^{-1}$ (avg) | 0.1 % $\pm$ 3 % RH (0–90 % RH), 0.1 % $\pm$ 5 % RH (90–100 % RH) | % |
| Precipitation | Ott Pluvio 2 v. 200 (heated) with wind shelter | Since Jul 2015 | 10 min (total) | 0.01 mm $\text{h}^{-1}$ $\pm$ 1 % | mm |
| | Vaisala WXT520[1] | Jul 2015 to Sep 2017 | 10 min (total) | 0.01 mm $\text{h}^{-1}$ $\pm$ 5 % | mm |
| Wind speed and direction | Kroneis 262 | Since Jul 2015 | 1 s 10 $\text{min}^{-1}$ (avg) | $\pm$ 0.2 m $\text{s}^{-1}$ (speed) | m $\text{s}^{-1}$ and ° |
| | Vaisala WXT520[1] | Jul 2015 to Sep 2017 | 1 s 10 $\text{min}^{-1}$ (avg) | 0.1 m $\text{s}^{-1}$ $\pm$ 3 % (speed) 1° $\pm$ 3 % for 10 m $\text{s}^{-1}$ (direction) | m $\text{s}^{-1}$ and ° |
| Wind gust and direction | Kroneis 262 | Since Jul 2015 (direction since Sep 2017) | 1 s 10 $\text{min}^{-1}$ (max) | $\pm$ 0.3 m $\text{s}^{-1}$ (speed) | m $\text{s}^{-1}$ and ° |
| | Vaisala WXT520[1] (gust speed only) | Jul 2015 to Sep 2017 | 1 s 10 $\text{min}^{-1}$ (max) | 0.1 m $\text{s}^{-1}$ $\pm$ 3 % (speed) | m $\text{s}^{-1}$ |
| Radiative fluxes (short- and longwave) | Kipp & Zonen CNR 4 (ventilated) | Since Jul 2015 | 1 min 10 $\text{min}^{-1}$ (avg) | 10-20 W $\text{m}^{-2}$ (incoming) 5-15 W $\text{m}^{-2}$ (outgoing) | W $\text{m}^{-2}$ W $\text{m}^{-2}$ |
| Atmospheric pressure | Young 61302V | Since Sep 2017 | 1 min 10 $\text{min}^{-1}$ (avg) | 0.2 hPa (25 °C), 0.3 hPa (-40 to 60 °C) | hPa |
| | Vaisala WXT520[1] | Jul 2015 to Sep 2017 | 1 min 10 $\text{min}^{-1}$ (avg) | 0.1 hPa $\pm$ 0.5 hPa (0 to 30 °C) 0.1 hPa $\pm$ 1.0 hPa (-52 to 60 °C) | hPa |
| Snow depth (exposed loc.) | Sommer USH-8 | Since Sep 2017 | 1 min 10 $\text{min}^{-1}$ (avg) | 1 mm $\pm$ 0.1 % | mm |
| Snow depth (sheltered loc.) | Sommer USH-9 | Since Sep 2020 | 1 min 10 $\text{min}^{-1}$ (avg) | 1 mm $\pm$ 0.1 % | mm |
| Snow water equivalent (exposed loc.) | Sommer snow pillow 3x3 | Since Sep 2017 | 1 min 10 $\text{min}^{-1}$ (avg) | unknown | mm w.e. |
| Snow water equivalent (sheltered loc.) | Sommer SSG-2 snow scale | Since Oct 2019 | 1 min 10 $\text{min}^{-1}$ (avg) | 0.1 mm w.e. $\pm$ 0.3 % | mm w.e. |
| Snow temperature profile (0, 20, 40, 60, 80, 100 cm, exposed loc.) | Sommer SCA snow temperature profile sensor | Since Sep 2017 | 1 min 10 $\text{min}^{-1}$ (avg) | $\pm$ 0.3 °C | °C |
| Snow temperature profile (0, 20, 40, 60, 80, 100 cm, sheltered loc.) | Sommer SCA snow temperature profile sensor | Since Sep 2017 | 1 min 10 $\text{min}^{-1}$ (avg) | $\pm$ 0.3 °C | °C |
| Snow drift | Sommer SND snow drift sensor | Since Sep 2020 | 10 min (total) 1 min 10 $\text{min}^{-1}$ (avg/min/max) | unknown | g $\text{m}^{-2}$ |

[1] Sensor discarded.

[2] Depending on air temperature; see technical data sheet of the manufacturer.

**Table 2.** Climate and snow variables recorded by the sensors installed at the station Latschbloder (2919 m a.s.l., 46.80106° N, 10.80659° E). Accuracy according to technical data sheets of the manufacturers. Temporal resolution of the data records is 10 min. The station is powered by solar panels and rechargeable battery packs.

| Variable | Sensor | Period of operation | Measurement interval | Resolution and accuracy | Unit |
|---|---|---|---|---|---|
| Air temperature | E+E EE08 (ventilated) | Since Sep 2017 | 1 min 10 min$^{-1}$ (avg) | < 0.5 °C[2] | °C |
| | Vaisala WXT520[1] | Sep 2013 to Sep 2017 | 1 min 10 min$^{-1}$ (avg) | 0.1 °C $\pm$ 0.3 °C | |
| Relative humidity | E+E EE08 | Since Sep 2017 | 1 min 10 min$^{-1}$ (avg) | $\pm$ 2 % RH (0–90 % RH), $\pm$ 3 % RH (90–100 % RH) | % |
| | Vaisala WXT520[1] | Sep 2013 to Sep 2017 | 1 min 10 min$^{-1}$ (avg) | 0.1 % $\pm$ 3 % RH (0–90 % RH), 0.1 % $\pm$ 5 % RH (90–100 % RH) | % |
| Precipitation | Ott Pluvio 2 v. 200 (unheated) with wind shelter | Since Jul 2014 | 10 min (total) | 0.01 mm h$^{-1}$ $\pm$ 1 % | mm |
| | Friedmann tipping bucket[1] | Sep 2013 to Jun 2014 | 10 min (total) | - | mm |
| | Vaisala WXT520[1] | Sep 2013 to Sep 2017 | 10 min (total) | 0.01 mm h$^{-1}$ $\pm$ 5 % | mm |
| Wind speed and direction | Young 05103 | Since Sep 2017 | 1 s 10 min$^{-1}$ (avg) | $\pm$ 0.3 m s$^{-1}$ (speed), $\pm$ 3 ° (direction) | m s$^{-1}$ and ° |
| | Vaisala WXT520[1] | Sep 2013 to Sep 2017 | 1 s 10 min$^{-1}$ (avg) | 0.1 m s$^{-1}$ $\pm$ 3 % (speed) 1° $\pm$ 3 % for 10 m s$^{-1}$ (direction) | m s$^{-1}$ and ° |
| Wind gust and direction | Young 05103 | Since Sep 2017 | 1 s 10 min$^{-1}$ (max) | $\pm$ 0.3 m s$^{-1}$ (speed), $\pm$ 3 ° (direction) | m s$^{-1}$ and ° |
| | Vaisala WXT520[1] (gust speed only) | Sep 2013 to Sep 2017 | 1 s 10 min$^{-1}$ (max) | 0.1 m s$^{-1}$ $\pm$ 3 % (speed) | m s$^{-1}$ |
| Radiative fluxes (short- and longwave) | Kipp & Zonen CNR 4 (ventilated) | Since Sep 2013 | 1 min 10 min$^{-1}$ (avg) | 10-20 W m$^{-2}$ (incoming) 5-15 W m$^{-2}$ (outgoing) | W m$^{-2}$ W m$^{-2}$ |
| Atmospheric pressure | Young 61302V | Since Sep 2017 | 1 min 10 min$^{-1}$ (avg) | 0.2 hPa (25 °C), 0.3 hPa (-40 to 60 °C) | hPa |
| | Vaisala WXT520[1] | Sep 2013 to Sep 2017 | 1 min 10 min$^{-1}$ (avg) | 0.1 hPa $\pm$ 0.5 hPa (0 to 30 °C) 0.1 hPa $\pm$ 1.0 hPa (-52 to 60 °C) | hPa |
| Snow depth | Sommer USH-8 | Since Sep 2017 | 1 min 10 min$^{-1}$ (avg) | 1 mm $\pm$ 0.1 % | mm |
| Snow temperature profile (0, 20, 40, 60, 80, 100 cm) | Sommer SCA snow temperature profile sensor | Since Sep 2020 | 1 min 10 min$^{-1}$ (avg) | $\pm$ 0.3 °C | °C |

[1] Sensor discarded.

[2] Depending on air temperature; see technical data sheet of the manufacturer.

**Table 3.** Climate and snow variables recorded by the sensors installed at the station Proviantdepot (2737 m a.s.l., 46.82951° N, 10.82407° E). Accuracy according to technical data sheets of the manufacturers. Temporal resolution of the data records is 10 min. The station is powered by solar panels and rechargeable battery packs.

| Variable | Sensor | Period of operation | Measurement interval | Resolution and accuracy | Unit |
|---|---|---|---|---|---|
| Air temperature | E+E EE08 (ventilated) | Since Oct 2019 | 1 min 10 min$^{-1}$ (avg) | < 0.5 °C[1] | °C |
| Relative humidity | E+E EE08 | Since Oct 2019 | 1 min 10 min$^{-1}$ (avg) | $\pm$ 2 % RH (0–90 % RH), $\pm$ 3 % RH (90–100 % RH) | % |
| Precipitation | Ott Pluvio 2 v. 200 (heated) with wind shelter | Since Oct 2019 | 10 min (total) | 0.01 mm h$^{-1}$ $\pm$ 1 % | mm |
| Wind speed and direction | Young 05103 | Since Oct 2019 | 1 s 10 min$^{-1}$ (avg) | $\pm$ 0.3 m s$^{-1}$ (speed), $\pm$ 3 ° (direction) | m s$^{-1}$ and ° |
| Wind gust and direction | Young 05103 | Since Oct 2019 | 1 s 10 min$^{-1}$ (max) | $\pm$ 0.3 m s$^{-1}$ (speed), $\pm$ 3 ° (direction) | m s$^{-1}$ and ° |
| Radiative fluxes (short- and longwave) | Kipp & Zonen CNR 4 (ventilated) | Since Oct 2019 | 1 min 10 min$^{-1}$ (avg) | 10-20 W m$^{-2}$ (incoming) 5-15 W m$^{-2}$ (outgoing) | W m$^{-2}$ W m$^{-2}$ |
| Atmospheric pressure | Young 61302V | Since Oct 2019 | 1 min 10 min$^{-1}$ (avg) | 0.2 hPa (25 °C), 0.3 hPa (-40 to 60 °C) | hPa |
| Surface temperature | Sommer SIR surface temperature sensor | Since Oct 2019 | 1 min 10 min$^{-1}$ (avg) | unknown | °C |
| Snow depth | Sommer USH-9 | Since Oct 2019 | 1 min 10 min$^{-1}$ (avg) | 1 mm $\pm$ 0.1 % | mm |
| Snow water equivalent | Sommer SSG-2 snow scale | Since Oct 2019 | 1 min 10 min$^{-1}$ (avg) | 0.1 mm w.e. $\pm$ 0.3 % | mm w.e. |
| Snow temperature profile (0, 20, 40, 60, 80, 100 cm) | Sommer SCA snow temperature profile sensor | Since Oct 2019 | 1 min 10 min$^{-1}$ (avg) | $\pm$ 0.3 °C | °C |
| Snow density | Sommer SPA-2 snow pack analyzer | Since Sep 2019 | 1 min 10 min$^{-1}$ (avg) | unknown | kg m$^{-3}$ |
| Snow liquid water content | Sommer SPA-2 snow pack analyzer | Since Sep 2019 | 1 min 10 min$^{-1}$ (avg) | unknown | vol % |
| Snow ice content | Sommer SPA-2 snow pack analyzer | Since Sep 2019 | 1 min 10 min$^{-1}$ (avg) | unknown | vol % |

[1]Depending on air temperature; see technical data sheet of the manufacturer.