# Peer review of "Operational and experimental snow observation systems in the upper Rofental: data from 2017 to 2023"

_Earth System Science Data, 2024_

## Author Response (AR1)

**Response to Referee #1**

In the following we give the point-by-point response to Referee #1 structured as follows:

(1) comments from the referee, (2) *our response,* and (3) *explanation of the changes in the revised manuscript. The given line numbers refer to the version of the manuscript with marked-up changes.*

The current version of the manuscript 'Operational and experimental snow observation systems in the upper Rofental: data from 2017 to 2023' describes a dataset collected at three sites in the Rofental in the Austrian Alps. The dataset contains standard measurements as well as some more experimental sensor setups for snow measurements, e.g., to record blowing snow events at one site.

The manuscript extends the ESSD paper 'The Rofental: a high Alpine research basin (1890–3770ma.s.l.) in the Oetztal Alps (Austria) with over 150 years of hydrometeorological and glaciological observations' by Strasser et al. 2018. Moreover, it was already submitted in a previous version to TDC in 2021 with a dataset until 2020, which was, however, at this time too short. This dataset was now extended with three more years, which makes the dataset in my opinion sufficiently long for publication in ESSD.

In general, the Rofental research catchment is a great site for high-alpine research. As such freely available datasets are still very scarse, this dataset can be very helpful for snow monitoring and the development and testing of snow models in high-alpine regions. The dataset is easily accessible and is fully described in the manuscript; data gaps are mentioned and discussed.

The manuscript in its current version has been significantly improved by the reviewing process in 2021 compared to its previous version (not published).

*Thank you very much for reviewing our manuscript! In the following we answer the questions and describe how we will improve the manuscript:*

I have only a few minor points:

p.7, l.160f: 'The station comprises a large set of operational snow cover sensors'. In this context, the word 'large' seems to be a bit too much. à 'The station comprises several operational snow cover sensors'

*We agree on this, and we will change the sentence accordingly.*

*We changed the sentence as suggested.*

p.10, Figure 4: Please indicate a line at the year 2017 as the dataset for this paper starts in this year.

*We will insert an indicating line in the Fig. 4.*

*We inserted a dashed line in Fig. 4 to indicate the installation of the snow sensors in September 2017 (as suggested also by Referee #3).*

p.11, Figure 13 (and further figures): It is good that you keep the same colour for individual stations throughout the manuscript. However, yellow might be a too light colour.

*We agree on this. We will revise the color schemes throughout the manuscript and avoid overly light colors.*

*We revised the color schemes for all figures in the manuscript.*

p.12, l. 262: Mention that the season 2019/2020 is an example in sub-section 7.2. Why do you not show further seasons?

*We will follow the suggestion and extend the analysis and include more seasons of snow temperature profiles in this section.*

*We extended the analysis and added a second season to Fig. 8. and the respective discussion in subsection 7.2, L. 331-337.*

p.17, Figure 11: I would choose other colours here than the colours, you chose for representing data of specific stations.

*We fully agree on this, and we will change the colors here so that they differ from the station color coding.*

*We followed the suggestion and used different color schemes for the figures not connected to the station color coding.*

p.18, l.369: '…over six winter seasons'. This is not the case for all sensors. Please add this information also in the conclusions.

*We will change the respective sentence accordingly.*

*We added the following sentence for clarification: "... over six winter seasons (September 2017 to August 2023). Data from the most recently installed station is available for four winter seasons (October 2019 to August 2023) and data from individual additional snow sensors for three seasons (September 2020 to August 2023) ."*

**Response to Referee #2**

In the following we give the point-by-point response to Referee #2 structured as follows:

(1) comments from the referee, (2) *our response,* and (3) *explanation of the changes in the revised manuscript. The given line numbers refer to the version of the manuscript with marked-up changes.*

This paper documents an excellent and important dataset for analyzing mountain snow processes. I highly recommend the publication of this paper and have only minor comments below. The only detractions I have in the score above on originality stem from the fact that some of this is an extension of a previous paper (albeit over a long enough period and with enough new data that it is well worth publishing), and on presentation quality there are a few english grammar changes that could be updated in the proof stage.

*Thank you very much for the thorough review of our manuscript! We answer the questions point by point in the following and explain the improvements we will make based on the respective issue/suggestion:*

1) In the context of alpine catchment hydrology, please document the vegetation cover (and lack there of) in most of this basin, there are trees and grasses at lower elevations, if not where these data were collected.

*We will add information about vegetation cover in the manuscript and include it in Fig. 1 (map of catchment and stations).*

*We added a landcover map to Fig. 1 and a respective explanation in Sect. 1., L. 84-86.*

2) Is there a flag provided in the data to note times when instruments are covered?

*The main instruments are rarely covered by snow. The wind and radiation sensors are sometimes covered or frozen (wind) for a short period. However, it is difficult to identify if - in each situation with suspicious measurements - the sensors are frozen, covered by snow, or if there is a logger problem. Therefore, values are just set to "missing value" in all these situations.*

3) Can you document how well the point observations of snow depth represent the surrounding snow depth at all sites? Clearly at the exposed/sheltered site pair, there is a lot of variability. Is that true at all sites?

*There is much less variability at the Proviantdepot and Latschbloder station compared to the two Bella Vista sites. The former two have been positioned to best represent their surroundings. However, at Latschbloder, a strong interannual variability of snow depth is discernible. It is generally difficult to quantify the variability as well as the representability of each station. However, we will add more description about how the stations are situated in the terrain and to what degree they should represent their surroundings regarding snow depth variability. We will add these explanations in Sects. 4 and 7.*

*We added explanation and discussion about the representativeness of the stations for their surroundings and the interannual variability to Sect. 7.9, L. 292-299 in the marked-up version of the revised manuscript.*

4) Are the 10-minute data an average of higher frequency measurements over the prior 10 minutes? The surrounding +/- 5 min period? or are they instantaneous?

*Yes, exactly. The 10-minute data are the result of high frequency measurements of the prior 10 minutes that are processed in the logger. Depending on the measured variable, the logger writes average (e.g. temperatures, rel. humidity, snow depth, SWE, radiation), total (precipitation) or maximum (wind gust). The higher frequency of measurement is 1 minute, except for wind speed and direction, where it is 1 second. We will add this information to the manuscript and to Tabs. 1,2, and 3.*

*We added this information to the manuscript in Sect. 5.1, L. 182-185, and to Tabs. 1,2, and 3.*

5) When replacing sensors, was any cross-validation/comparison made between old and new sensors?

*A validation of the new sensors was performed by the manufacturer before installation. The old sensors were replaced by the new ones, so we didn't do such a comparison on site.*

6) What is done as part of the "thorough check for obvious errors"? Is this primarily thresholds and change thresholds? Are the "corrected" values flagged as such?

*Exactly, the thorough check is based on thresholds and on (non)-change thresholds (e.g. wind speeds staying constant (frozen sensor), rel. hum. >100%, etc.). The respective time steps are not "corrected" but set to and marked as "missing value". We will add more details on this in Sect. 5.1.*

*We added some more detail on this in Sect. 5.1, L. 189-190 in the marked-up version of the revised manuscript.*

I enjoyed the evaluation of more experimental measurements of snow flux, snow density, and drift occurence. The authors might want to look at a similar dataset that was recently published and included snow particle flux as well as ~5-minute repeat scans with terrestrial scanning lidar during many such blowing snow events (Lundquist et al 2024 https://doi.org/10.1175/BAMS-D-23-0191.1)

*We will include a reference to the very interesting publication and data in the introduction and Sect. 7.*

*We now cite the data collection in the introduction (L.61-62) and the part of the study about LIDAR scanning of blowing snow events in Sect. 7.4, L. 434-436 in the revised manuscript.*

**Response to Referee #3**

In the following we give the point-by-point response to Referee #3 structured as follows:

(1) comments from the referee, (2) *our response,* and (3) *explanation of the changes in the revised manuscript. The given line numbers refer to the version of the manuscript with marked-up changes.*

This manuscript presents a comprehensive data set for three research sites in the Upper Rofental. The time series presented comprise data from 2017 to 2023 and come from three meteorological and snow-hydrological stations. The manuscript extends a ESSD paper published in 2018. Since 2017, the observation network has been extended with a new automatic weather station and has been complemented by sensors continuously recording snow cover properties. The manuscript documents these extensions and presents the new data sets that have been recorded between 2017 and 2023.

The Rofental research catchment is a very well-instrumented high Alpine environmental research basin, combining glaciological, hydrological, and meteorological observations. The dataset presented here is very valuable, especially concerning new sensor technologies measuring snow properties and certainly merits publication in ESSD. The dataset is easy to access and well documented.

The manuscript presents interesting exemplary use cases of data analysis, but I think several points must be clarified:

*We thank you very much for the thorough review of our manuscript and the valuable suggestions! We answer your questions in the following and describe how we will improve the manuscript accordingly.*

Line 174: 'temperature correction for longwave radiation…', specify sensor temperature to correct for the longwave emission of the sensor (I suppose)

*Yes, we missed to specify the procedure precisely. We will change the sentence as suggested.*

*We changed the sentence to "… correction of long-wave radiation for sensor temperature…" (L. 187).*

Section 6 'Meteorological data'

Is the wind speed following seasonal changes?

*Indeed, wind speed shows a seasonal pattern with higher values occurring in the winter months. This reflects the general situation of a higher storm frequency in winter caused by frequent distinct low-pressure systems and Foehn events. We will add an explanation in Sect. 6.*

*We added a respective explanation to the manuscript (Sect. 6, L. 232-234 in the revised marked-up manuscript).*

Specify how wind gusts are measured, what does 'wind gust' mean?

*Wind gust means the maximum wind speed measured in a certain interval (here within 10 minutes). It is measured by logging the maximum wind speed in the 10 min. interval (prior and including the time stamp) from measurements performed in a resolution of 1 second, as opposed to the average when referring to average wind speed. We will clarify this in the manuscript in Sects. 5.1 and 6.*

*We added this explanation to Sect. 5.1, L. 182-185.*

Rather than discussing outgoing solar radiation, present and discuss albedo values.

*Correctly calculating albedo from incoming and outgoing solar radiation needs careful assessment of the terrain and solar incident angles to perform a slope and sensor tilt correction. This entails its own uncertainties specifically when radiation values are very low. We are generally not presenting secondary analysis of further processed data in this manuscript but focus on providing the measurements. We would leave that as a potential application to users of the data.*

The precipitation gauge under-catch is a serious measurement problem and may explain the maximal amounts of precipitation recorded in summer. Further analysis is required. The under-catch can be quantified (at least approximately) by comparing the measurements from different types of rain gauges and the snow scale in winter.

*We fully agree on this. However, this is a common issue with precipitation measurements – specifically in high-mountain regions - and many investigations and correction methods exist in the literature. We focus on providing the data and potential users can perform further analyses depending on their use case. However, as suggested, we investigated the under-catch issue in the data for the Proviantdepot station. The table below shows precipitation totals in mm for the four winter seasons 2019 – 2023 (each Oct to Jun) and maximum SWE on the snow scale. Precipitation is filtered using threshold air temperatures 1°C and 2°C to assess snowfall totals. This simple approximation includes uncertainties such as intermittent SWE decrease, uncertainties in phase separation by threshold temperatures, precipitation under-catch, wind-driven redistribution on/off the snow scale, and the small-scale variability between precipitation gauge and snow scale. Results show a strong variability between the four winter seasons. The strong deviation in the first two winter seasons is probably a combination of deposition of wind-blown snow on the scale (see description in the manuscript Sect. 7.2 and Fig. 8 / comment below) and at the same time under-catch at the gauge. We will add a respective explanation to the manuscript.*

|  | Oct 19 – Jun 20 | Oct 20 – Jun 21 | Oct 21 – Jun 22 | Oct 22 – Jun 23 |
|---|---|---|---|---|
| Precip. mm | 685.47 | 417.73 | 551.04 | 503.96 |
| Precip mm (T <= 1°C) | 593.92 | 317.21 | 337.98 | 390.61 |
| Precip mm (T <= 2°C) | 633.01 | 337.05 | 366.76 | 426.27 |
| Max. SWE mm w.e. | 927.5 | 792.6 | 453.1 | 429.5 |

*We added a statement about the undercatch and its strong interannual variability to the manuscript (Sect. 6, L. 249-251 in the revised version).*

Lines 248-249: No explanation for the differences in HS measurements a few meters apart from January to May 2022 (panel (e))?

*This was most probably caused by very small-scale wind-driven snow redistribution. We could clearly observe such a case in the season 2019/2020 on webcam images where the melt out date obtained by the SWE scale was significantly delayed compared to the HS measurement (see Sect. 7.2 and Fig. 8). In the season 2021/2022 however, the melt out date is almost identical despite the differences in HS. Therefore, we cannot clearly show it in the webcam for that season. We will add further clarification and explanation on this to the manuscript.*

*We added a respective clarification to the manuscript in Sect. 7.1, L. 271-277. It is also explained in Sect. 7.2, L. 312-318*

Section 7.2 and Figure 8 are not clear.

The mismatch between SWE and HS measurements is worrying and should be analyzed further. As the sensors used are not sufficiently specified (see other comments), it is difficult to understand if the measured variables are at the same site and the interpretation of the data remains unclear. For instance, the legend of Figure 8 mentions 'SWE melt out date is two weeks later than HS', but we don't not if these variables are measured at the same location. Furthermore, temperature measurements in panel (b) should be used to estimate the melt out date (before HS melt out date?). The different temperatures in panel (c) are not visible (use different colors).

*We agree on this, and we will extend the given explanation for the mismatch including a better specification of the exact sensors and locations. We will also present additional seasons here to show that the mismatch is not a persistent pattern and strongly varies interannually (see also Figs. 6 and 7). In addition, we will change the color scheme of the temperature panel so that melt out date is clearly visible in the 0 cm and surface temperature measurements.*

*Please see the response to the previous comment. We extended the explanation on this in the revised manuscript (Sect. 7.1, L. 271-277 and Sect. 7.2, L. 312-318 and in the caption of Fig. 8). We changed the color scheme in the Fig. 8 and filtered temperature data with measured HS for the respective sensor height (explained in the caption of Fig. 8).*

Section 7.3

The two configurations of the flat bands (in diagonal or horizontally) should be described before (in Section 4.3).

*Yes, we will add the description of the flat bands configuration to Sect. 4.3.*

*We added a description of the flat band configuration to Sect. 4.3 (L. 169-172 in the revised marked-up manuscript).*

The comparison of snow density values derived from S1, S2 and the snow scale is unclear (lines 314-321). Should the density derived from the snow scale (total SWE and HS) only be compared with the S1 diagonal flat band measurements, since the S2 measurements concern

snow density at the base of the snowpack? The interpretation of density measurements needs to be clarified.

*Yes, for a direct quantitative evaluation it makes more sense to compare the HS/SWE derived density (average for the whole snowpack) to the diagonal band. However, this holds true only for the case where HS does not exceed the height of the diagonal band and the band is sufficiently buried. Therefore, we decided to show the density of both flat bands (diagonal and base of snowpack) in the figure to see the differences. We will add a clarifying discussion to the manuscript.*

*We added the following statement on this in Sect. 7.3, L. 369-371: "It is noted that the snow scale measurements of SWE and snow density are bulk values for the whole snowpack independent of HS while the SPA measures properties only for the layer around the respective flat band, i.e., the base layer of the snow pack for S2, and the layer that is spanned by the diagonal band for S1, respectively."*

Section 7.4

Lines 326-328: 'The measurement principle… with different results'. As the uncertainty of acoustic snow drift sensor seems quite high, could the authors be more specific? Give an estimation of the uncertainty range, the main measurement problems… The following sentence states that 'it is still the only way to continuously measure and detect drifting snow events with a certain reliability'. What about optical snow particle counters?

*We will further elaborate on the uncertainties and measurement problems of the acoustic snow drift sensor. We will also add optical snow particle counters to the discussion here.*

*We extended the literature review and added optical snow particle counters (Sect. 7.4, L. 378-384).*

The analysis of a snow drift event based on different measurement instruments is interesting but I see two main problems:

- the SWE is measured with different types of sensors in the exposed and sheltered sites (snow pillow and snow scale). As SWE measurements by different sensors can be quite different (for instance Figure 7), the analyze of SWE differences between the exposed and sheltered sites require a better comparison between snow pillow and snow scale measurements (a comparison at the same location for instance).

*Unfortunately, we do not have a side-by-side setup of snow pillow and scale and therefore we cannot perform such a comparison. We are planning to replace the snow pillow by a scale as soon as the funding is available. We will add a statement to the added uncertainties induced by different types of SWE sensors to the manuscript.*

*We inserted a note about the potentially added uncertainties by using two different sensor types in Sect. 7.1, L. 279-281.*

- The blowing snow flux measured by acoustic sensor can be perturbed by snowfall. Thus, with this sensor (and due to the difficulty to measure the snowfall rate in strong wind conditions), it is difficult to quantify the blowing snow flux during a precipitation event and

to relate it to a wind speed threshold for snow erosion. Thus, the interpretation of snow particle fluxes and changes in SWE is problematic. It would be more convincing to analyze a snow drift event without precipitation.

*We agree on the added difficulty of flux quantification during a snowfall event. We will look for an event without precipitation and subsequently analyze and present it here.*

*We included another period with two blowing snow events without precipitation (Fig. 12) and added the respective results and discussions to Sect. 7.4, L. 409-438.*

In panel (a), snow depth in the sheltered site shows little deposition compared with the large deposition recorded in SWE (panel (a)). This shows a discrepancy between SWE and HS measurements at the sheltered site during the period of the main blowing snow event?

*Yes, this observation shows the large heterogeneity even at a very small scale for the same station location. However, we will further investigate on the persistence of this issue when we look for more blowing snow events (see comment above) and add a respective explanation to the manuscript.*

*Please see comment above, we discuss more blowing snow events and see similar inconsistencies. The small-scale deviations of related HS and SWE measurements are also further discussed and explained in Sect. 7.1.*

Figures:

The text in the figures is often too small and difficult to read (Figures 4, 5, 6, 7, 8, 9 and 11, axe titles in particular). The legend should clearly state from which sensor is derived each variable (for instance from which sensor is derived precipitation in Figure 5 or 11?). This is an important point.

*We will revise the figures to enhance readability and enlarge the axis titles. The presented data are all from the most recent sensors. We will state this clearly in the manuscript and highlight the respective sensors in Tabs. 1, 2, and 3.*

*We revised all figures in the manuscript. We adjusted the color schemes to avoid too bright colors and enhanced readability by increasing font sizes and line widths. We now clearly state in the manuscript that alle the data shown is from the most recent sensors (Sect. 6, L. 224-225). We also marked discarded sensors in Tabs. 1, 2, and 3, so that it is easily visible for each variable that data from Sep 2017 to Aug 2023 can only be retrieved from the most recent sensors.*

The map in figure 1 should clearly highlight the three stations discussed in the paper.

*We will revise the map and highlight the three stations.*

*We completely revised the map in Fig. 1 and highlighted the three stations. We also added information on land cover (as suggested by Referee #2).*

Figure 3 is very useful but should clearly highlight the new instruments (compared to the 2018 publication).

*We will highlight new instruments compared to the 2018 publication in Fig. 3 and in Tabs. 1, 2, and 3.*

*We added a vertical line in Fig. 4 (as suggested by Referee #1) to highlight the sensors and data period presented in this manuscript. We marked the discarded sensors in Tabs. 1,2, and 3 instead of highlighting the new ones because we also discuss data from previously existing sensors for the new period 2017-2023 here. We decided not to mark the new sensors in Fig. 3 because it should give a schematic overview of all available instruments and show the differences between the three stations.*

Legend of Figure 4 'Narrow bars indicate a second sensor for a variable': not clear to me.

*We agree that this is not clear. We will add more explanation in the caption and add information in the figure legend.*

*We added the following explanation to the caption of Fig. 4: "Some variables are recorded with two different sensors at the same location. These are indicated by (2x) and the narrow bars show the data availability for the second sensor."*

Figure 6: SWE in mm w.e. The scale of HS should go to 200 cm in (c) to be coherent with (b) and (e). In (e): 'USH-9' is not clear.

*We will change the unit of SWE to "mm w.e." throughout the manuscript. We will change the scale to a coherent value of 200 cm and add explanation for the presented sensor (USH-9) in the legend and in the caption.*

*We changed the unit of SWE to "mm w.e." throughout the manuscript and adjusted the range of the y-axis in Fig. c) to be coherent with b) and e). We added explanation for the HS sensors at the Proviantdepot station in Sect. 7.1, L. 269-273.*

Figure 7 is interesting but not clear. Specify from which the sensors are derived HS and SWE. The yellow line is not sufficiently visible (chose another color).

*Fig. 7 shows all available HS and SWE measurements. We will clarify this in the caption and in the Sect. 7.1. Furthermore, we will revise the color schemes for all figures to enhance readability and we will avoid the too bright yellow color.*

*We added a more detailed explanation and extended the discussion about the different measurements of HS and SWE in Sect. 7.1, L. 269-277 and L. 292-299. We also revised the color schemes of all figures to avoid too bright colors and enhanced readability by adjusting font sizes and line widths.*

Figure 9: explain S1 and S2 in the legend. Panel (c): SWE in mm w.e.

*We will add the specification of S1 and S2 in the legend and we will change the unit of SWE to "mm w.e." throughout the manuscript.*

*We changed the unit of SWE to "mm w.e." throughout the manuscript and added the specification of S1 and S2 to the caption of Fig. 9.*

Tables 1, 2 and 3: better highlight the new sensors installed since 2018

*We will highlight the new sensors in the tables.*

*See comment above: we decided to mark the discarded sensors in Tabs. 1,2, and 3 instead of highlighting the new ones because we show data from previously existing sensors for the new period 2017-2023 here as well. In addition, we added a vertical line in Fig. 4 (as suggested by Referee #1) to highlight the sensors and data period presented in this manuscript.*